# The genome of the Hi5 germ cell line from *Trichoplusia ni*, an agricultural pest and novel model for small RNA biology

Yu Fu[1,2], Yujing Yang[3†], Han Zhang[3], Gwen Farley[3], Junling Wang[3], Kaycee A Quarles[3], Zhiping Weng[2*], Phillip D Zamore[3*]

[1]Bioinformatics Program, Boston University, Boston, United States; [2]Program in Bioinformatics and Integrative Biology, University of Massachusetts Medical School, Worcester, United States; [3]RNA Therapeutics Institute and Howard Hughes Medical Institute, University of Massachusetts Medical School, Worcester, United States

**\*For correspondence:**
zhiping.weng@umassmed.edu (ZW);
phillip.zamore@umassmed.edu (PDZ)

**Present address:** [†]Laboratory of Pathology, State Key Laboratory of Biotherapy and Department of Pathology, West China Hospital, West China Medical School, Sichuan University, Chengdu, China

**Competing interests:** The authors declare that no competing interests exist.

**Abstract** We report a draft assembly of the genome of Hi5 cells from the lepidopteran insect pest, *Trichoplusia ni*, assigning 90.6% of bases to one of 28 chromosomes and predicting 14,037 protein-coding genes. Chemoreception and detoxification gene families reveal *T. ni*-specific gene expansions that may explain its widespread distribution and rapid adaptation to insecticides. Transcriptome and small RNA data from thorax, ovary, testis, and the germline-derived Hi5 cell line show distinct expression profiles for 295 microRNA- and >393 piRNA-producing loci, as well as 39 genes encoding small RNA pathway proteins. Nearly all of the W chromosome is devoted to piRNA production, and *T. ni* siRNAs are not 2´-*O*-methylated. To enable use of Hi5 cells as a model system, we have established genome editing and single-cell cloning protocols. The *T. ni* genome provides insights into pest control and allows Hi5 cells to become a new tool for studying small RNAs ex vivo.
DOI: https://doi.org/10.7554/eLife.31628.001

## Introduction

Lepidoptera (moths and butterflies), one of the most species-rich orders of insects, comprises more than 170,000 known species (*Mallet, 2007*; *Chapman, 2009*), including many agricultural pests. One of the largest lepidopteran families, the Noctuidae diverged over 100 million years ago (mya) from the Bombycidae—best-known for the silkworm, *Bombyx mori* (*Rainford et al., 2014*). The Noctuidae family member cabbage looper (*Trichoplusia ni*) is a widely distributed generalist pest that feeds on cruciferous crops such as broccoli, cabbage, and cauliflower (*Capinera, 2001*). *T. ni* has evolved resistance to the chemical insecticide Dichlorodiphenyltrichloroethane (DDT; (*McEwen and Hervey, 1956*) and the biological insecticide *Bacillus thuringiensis* toxin (*Janmaat and Myers, 2003*), rendering pest control increasingly difficult. A molecular understanding of insecticide resistance requires a high-quality *T. ni* genome and transcriptome.

Hi5 cells derive from *T. ni* ovarian germ cells (*Granados et al., 1986*; *1994*). Hi5 cells are a mainstay of recombinant protein production using baculoviral vectors (*Wickham et al., 1992*) and hold promise for the commercial-scale production of recombinant adeno-associated virus for human gene therapy (*Kotin, 2011*; *van Oers et al., 2015*). Hi5 cells produce abundant microRNAs (miRNAs) miRNAs, small interfering RNAs (siRNAs), and PIWI-interacting RNAs (*Kawaoka et al., 2009*) (piRNAs), making them one of just a few cell lines suitable for the study of all three types of animal small RNAs. The most diverse class of small RNAs, piRNAs protect the genome of animal reproductive cells by silencing transposons (*Saito et al., 2006*; *Vagin et al., 2006*; *Brennecke et al., 2007*; *Houwing et al., 2007*; *Aravin et al., 2007*; *Kawaoka et al., 2008*). The piRNA pathway has been extensively studied in the dipteran insect

**eLife digest** A common moth called the cabbage looper is becoming increasingly relevant to the scientific community. Its caterpillars are a serious threat to cabbage, broccoli and cauliflower crops, and they have started to resist the pesticides normally used to control them. Moreover, the insect's germline cells – the ones that will produce sperm and eggs – are used in laboratories as 'factories' to artificially produce proteins of interest.

The germline cells also host a group of genetic mechanisms called RNA silencing. One of these processes is known as piRNA, and it protects the genome against 'jumping genes'. These genetic elements can cause mutations by moving from place to place in the DNA: in germline cells, piRNA suppresses them before the genetic information is transmitted to the next generation. Not all germline cells grow equally well under experimental conditions, or are easy to use to examine piRNA mechanisms in a laboratory. The germline cells from the cabbage looper, on the other hand, have certain characteristics that would make them ideal to study piRNA in insects.

However, the genome of the moth had not yet been fully resolved. This hinders research on new ways of controlling the pest, on how to use the germline cells to produce more useful proteins, or on piRNA.

Decoding a genome requires several steps. First, the entire genetic information is broken in short sections that can then be deciphered. Next, these segments need to be 'assembled' – put together, and in the right order, to reconstitute the entire genome. Certain portions of the genome, which are formed of repeats of the same sections, can be difficult to assemble. Finally, the genome must be annotated: the different regions – such as the genes – need to be identified and labeled.

Here, Fu et al. assembled and annotated the genome of the cabbage looper, and in the process developed strategies that could be used for other species with a lot of repeated sequences in their genomes. Having access to the looper's full genetic information makes it possible to use their germline cells to produce new types of proteins, for example for pharmaceutical purposes. Fu et al. went on to make working with these cells even easier by refining protocols so that modern research techniques, such as the gene-editing technology CRISPR-Cas9, can be used on the looper germline cells.

The mapping of the genome also revealed that the genes involved in removing toxins from the insects' bodies are rapidly evolving, which may explain why the moths readily become resistant to insecticides. This knowledge could help finding new ways of controlling the pest.

Finally, the genes involved in RNA silencing were labeled: results show that an entire chromosome is the source of piRNAs. Combined with the new protocols developed by Fu et al., this could make cabbage looper germline cells the default option for any research into the piRNA mechanism. How piRNA works in the moth could inform work on human piRNA, as these processes are highly similar across the animal kingdom.

DOI: https://doi.org/10.7554/eLife.31628.002

*Drosophila melanogaster* (fruit fly), but no piRNA-producing, cultured cell lines exist for dipteran germline cells. *T. ni* Hi5 cells grow rapidly without added hemolymph (*Hink, 1970*), are readily transfected, and—unlike *B. mori* BmN4 cells (*Iwanaga et al., 2014*), which also express germline piRNAs—remain homogeneously undifferentiated even after prolonged culture. In contrast to *B. mori*, no *T. ni* genome sequence is available, limiting the utility of Hi5 cells.

To further understand this agricultural pest and its Hi5 cell line, we combined divers genomic sequencing data to assemble a chromosome-level, high-quality *T. ni* genome. Half the genome sequence resides in scaffolds > 14.2 megabases (Mb), and >90% is assembled into 28 chromosome-length scaffolds. Automated gene prediction and subsequent manual curation, aided by extensive RNA-seq data, allowed us to examine gene orthology, gene families such as detoxification proteins, sex determination genes, and the miRNA, siRNA, and piRNA pathways. Our data allowed assembly of the gene-poor, repeat-rich W chromosome, which remarkably produces piRNAs across most of its length. To enable the use of cultured *T. ni* Hi5 cells as a novel insect model system, we established methods for efficient genome editing using the CRISPR/Cas9 system (*Ran et al., 2013*) as well as single-cell cloning. With these new tools, *T. ni* promises to become a powerful companion to flies to

study gene expression, small RNA biogenesis and function, and mechanisms of insecticide resistance in vivo and in cultured cells.

## Results

### Genome sequencing and assembly

We combined Pacific Biosciences long reads and Illumina short reads (*Figure 1A*, *Table 1*, and Materials and methods) to sequence genomic DNA from Hi5 cells and *T. ni* male and female pupae. The initial genome assembly from long reads (46.4 × coverage with reads >5 kb) was polished using paired-end (172.7 × coverage) and mate-pair reads (172.0 × coverage) to generate 1976 contigs spanning 368.2 megabases (Mb). Half of genomic bases reside in contigs > 621.9 kb (N50). Hi-C long-range scaffolding (186.5 × coverage) produced 1031 scaffolds (N50 = 14.2 Mb), with >90% of the sequences assembled into 28 major scaffolds. Karyotyping of metaphase Hi5 cells revealed that these cells have 112 ± 5 chromosomes (*Figure 1B*, *Figure 1—figure supplement 1*). Because lepidopteran cell lines are typically tetraploid (*Hink, 1972*), we conclude that the ~368.2 Mb *T. ni* genome comprises 28 chromosomes: 26 autosomes plus W and Z sex chromosomes (see below).

To evaluate the completeness of the assembled *T. ni* genome, we compared it to the Arthropoda data set of the Benchmark of Universal Single-Copy Orthologs (*Simão et al., 2015*) (BUSCO v3). The *T. ni* genome assembly captures 97.5% of these gene orthologs, more than either the silkworm (95.5%) or monarch butterfly (*D. plexippus*; 97.0%) genomes (*Supplementary file 1A*). All 79 ribosomal proteins conserved between mammals and *D. melanogaster* (*Yoshihama et al., 2002*; *Marygold et al., 2007*) have orthologs in *T. ni*, further evidence of the completeness of the genome assembly (*Supplementary file 1B*). Finally, a search for genes in the highly conserved nuclear oxidative phosphorylation (OXPHOS) pathway (*Porcelli et al., 2007*) uncovered *T. ni* orthologs for all known *D. melanogaster* OXPHOS genes (*Supplementary file 1C*).

The genomes of wild insect populations are typically highly heterogeneous, which poses a significant impediment to assembly (*Keeling et al., 2013*; *You et al., 2013*). We were unable to generate an isogenic *T. ni* strain by inbreeding. Therefore, our *T. ni* sequence reflects the genome of Hi5 cells, not cabbage looper itself. Hi5 cells presumably derive from a single immortalized, germline founder cell, which should reduce genomic variation among the cell line's four sets of chromosomes. To test this supposition, we identified the sequence variants in the Hi5 genome. In total, we called variants at 165,370 genomic positions (0.0449% of the genome assembly), with 2710 in predicted coding regions (0.0132% of coding sequence), indicating that the genome of Hi5 cells is fairly homogenous. For the majority (88.8%) of these genomic positions (covering 0.0399% of the genome), only one copy of the chromosome has the variant allele while the other three chromosomal copies match the reference genome. We can make three conclusions. First, Hi5 cells originated from a single founder cell or a homogenous population of cells. Second, the founder cells were haploid. Third, most sequence variants were acquired after the original derivation of the line from *T. ni* eggs.

We also assembled de novo *T. ni* genomes using paired-end DNA-seq data obtained from male and female pupae, but the resulting assemblies are fragmented (scaffold N50 ≤ 2.4 kb, *Supplementary file 1D*), likely due to the limitations of short-insert libraries and the high levels of heterozygosity commonly observed for genomes of wild insect populations (*Keeling et al., 2013*; *You et al., 2013*). The animal genome contigs are highly concordant with the Hi5 genome, with ≤1.37% of animal contigs misassembled (*Supplementary file 1D*). Although we cannot determine scaffold-level differences between the animal and Hi5 cells, at the contig-level the Hi5 genome assembly is representative of the *T. ni* animal genome.

### Gene orthology

We annotated 14,034 protein-coding genes in the *T. ni* genome (*Supplementary file 1E*), similar to other Lepidoptera (*Challis et al., 2016*). Analysis of the homology of *T. ni* genes to genes in 20 species that span the four common insect orders (Lepidoptera, Diptera, Coleoptera, Hymenoptera), non-insect arthropods, and mammals defines 30,448 orthology groups each containing orthologous proteins from two or more species (*Hirose and Manley, 1997*); 9112 groups contain at least one *T. ni* gene. In all, 10,936 *T. ni* protein-coding genes are orthologous to at least one gene among the 20 reference species (*Figure 1C*, *Figure 1—figure supplement 2*).

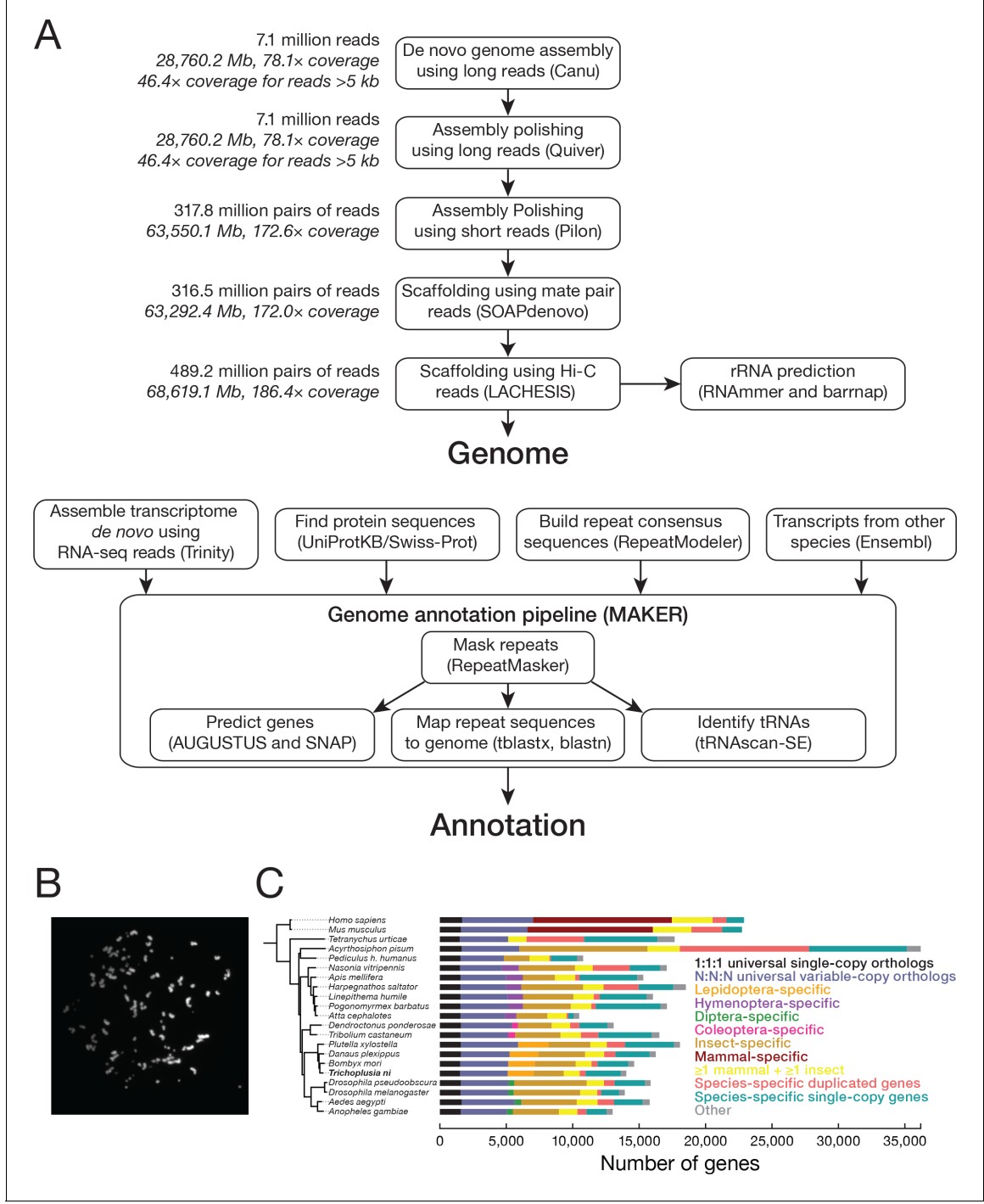

**Figure 1.** Chromosomes and genes in the *T. ni* genome based on data from the Hi5 cell line. (**A**) Genome assembly and annotation workflow. (**B**) An example of a DAPI-stained spread of Hi5 cell mitotic chromosomes used to determine the karyotype. (**C**) Phylogenetic tree and orthology assignment of *T. ni* with 18 arthropod and two mammalian genomes. Colors denote gene categories. The category 1:1:1 represents universal single-copy orthologs, allowing absence and/or duplication in one genome. N:N:N orthologs include orthologs with variable copy numbers across species, allowing absence in one genome or two genomes from different orders. Lepidoptera-specific genes are present in at least three of the four lepidopteran genomes; Hymenoptera-specific genes are present in at least one wasp or bee genome and at least one ant genome. Coleoptera-specific genes are present in both coleopteran genomes; Diptera-specific genes present in at least one fly genome and one mosquito genome. Insect indicates other insect-specific genes. Mammal-specific genes are present in both mammalian genomes. The phylogenetic tree is based on the alignment of 1:1:1 orthologs.
DOI: https://doi.org/10.7554/eLife.31628.003

The following figure supplements are available for figure 1:

*Figure 1 continued on next page*

*Figure 1 continued*

**Figure supplement 1.** Hi5 cell Karyotyping.

DOI: https://doi.org/10.7554/eLife.31628.004

**Figure supplement 2.** Phylogenetic tree of 21 species showing the scale, branch lengths and bootstrap support.

DOI: https://doi.org/10.7554/eLife.31628.005

**Figure supplement 3.** Opsins in insects.

DOI: https://doi.org/10.7554/eLife.31628.006

*T. ni* contains 2,287 Lepidoptera-specific orthology groups (*T. ni*, *B. mori*, *D. plexippus*, and *P. xylostella* [diamondback moth]). Far fewer orthology groups are unique to Diptera (404), Coleoptera (371), or Hymenoptera (1344), suggesting that the lepidopteran lifestyle requires more order-specific genes. The *T. ni* genome additionally contains 3098 orphan protein-coding genes for which we could detect no orthologous sequences in the 20 reference species. Of these orphan genes, 14.5% are present as two or more copies in the genome ('in-paralogs'), suggesting they evolved recently. Some of these in-paralogs may have arisen by gene duplication after the divergence of *T. ni* and *B.*

**Table 1.** Genome and gene set statistics for *T. ni* and *B. mori* (**International Silkworm Genome Consortium, 2008**). Cytochrome P450s, glutathione S-transferases, carboxylesterases, and ATP-binding cassette transporters for *B. mori* were retrieved from (**Yu et al., 2008**; **Yu et al., 2009**; **Ai et al., 2011**; **Liu et al., 2011b**).

| | *T. ni* | *B. mori* |
|---|---|---|
| Genome metrics | | |
| Genome size (Mb) | 368.2 | 431.7 |
| Chromosome count | 28 | 28 |
| Scaffold N50 (Mb) | 14.2 | 3.7 |
| Contig N50 (kb) | 621.9 | 15.5 |
| Mitochondrial genome (kb) | 15.8 | 15.7 |
| Quality control metrics | | |
| BUSCO complete (%) | 97.5 | 95.5 |
| CRP genes (%) | 100% | 100% |
| OXPHOS genes (%) | 100% | 100% |
| Genomic features | | |
| Repeat content (%) | 20.5% | 43.6% |
| GC content | 35.6% | 37.3% |
| CpG (O/E) | 1.07 | 1.13 |
| Coding (%) | 5.58 | 4.11 |
| Sex chromosomes | ZW | ZW |
| Gene statistics | | |
| Protein-coding genes | 14,043 | 14,623 |
| with Pfam matches | 9295 | 9685 |
| with GO terms | 9790 | 10,148 |
| Cytochrome P450 proteins | 108 | 83 |
| Glutathione S-transferases | 34 | 23 |
| Carboxylesterases | 87 | 76 |
| ATP-binding cassette transporters | 54 | 51 |
| Universal orthologs lost | 156 | 75 |
| Species-specific genes | 3098 | 2313 |

DOI: https://doi.org/10.7554/eLife.31628.007

*mori* ~111 mya (*Gaunt and Miles, 2002*; *Rota-Stabelli et al., 2013*; *Wheat and Wahlberg, 2013*; *Rainford et al., 2014*).

## Opsins

The ability of insects to respond to light is crucial to their survival. Opsins, members of the G-protein-coupled receptor superfamily, play important roles in vision. Covalently bound to light-sensing chromophores, opsins absorb photons and activate the downstream visual transduction cascade (*Terakita, 2005*). The *T. ni* genome encodes ultraviolet, blue, and long-wavelength opsins. Thus, this nocturnal insect retains the full repertoire of insect opsins and has color vision (*Zimyanin et al., 2008*) (*Figure 1—figure supplement 3*). *T. ni* also encodes an ortholog of the non-visual Rh7 opsin, which is found in a variety of insects (*International Glossina Genome Initiative, 2014*; *Futahashi et al., 2015*). In the *D. melanogaster* brain, Rh7 opsin participates in the entrainment of circadian rhythms by sensing violet light (*Ni et al., 2017*). *T. ni* also encodes an ortholog of the vertebrate-like opsin, pterosin, which was first detected in the honeybee (*A. mellifera*) brain and is found widely among insects except for *Drosophilid* flies (*Velarde et al., 2005*).

## Sex determination

Understanding the *T. ni* sex-determination pathway holds promise for engineering sterile animals for pest management. ZW and ZO chromosome systems determine sex in lepidopterans: males are ZZ and females are either ZW or ZO (*Traut et al., 2007*). To determine which system *T. ni* uses and to identify which contigs belong to the sex chromosomes, we sequenced genomic DNA from male and female pupae and calculated the male:female coverage ratio for each contig. We found that 175 presumably Z-linked contigs (20.0 Mb) had approximately twice the coverage in male compared to female DNA (median male:female ratio = 1.92; *Figure 2A*, *Figure 2—figure supplement 1A*). Another 276 contigs (11.1 Mb) had low coverage in males (median male:female ratio = 0.111), suggesting they are W-linked. We conclude that sex is determined in *T. ni* by a ZW system in which males are homogametic (ZZ) and females are heterogametic (ZW).

For some lepidopteran species, dosage compensation has been reported to equalize Z-linked transcript abundance between ZW females and ZZ males in the soma, while other species show higher expression of Z-linked genes in males (*Walters et al., 2015*; *Gu et al., 2017*). In the soma, *T. ni* compensates for Z chromosome dosage: transcripts from Z-linked genes are approximately equal in male and female thoraces (Z ≈ ZZ, *Figure 2B*). In theory, somatic dosage compensation could reflect increased transcription of the single female Z chromosome, reduced transcription of both male Z chromosomes, or silencing of one of the two male Z chromosomes.

To distinguish among these possibilities, we compared the abundance of Z-linked and autosomal transcripts (Z/AA in female and ZZ/AA in male, *Figure 2—figure supplement 1B and C*). Z-linked transcripts in the male thorax are expressed at lower levels than autosomal transcripts, but not as low as half (ZZ ≈ 70% AA). These data support a dosage compensation mechanism that decreases transcription from each Z chromosome in the *T. ni* male soma, but does not fully equalize Z-linked transcript levels between the sexes (Z ≈ ZZ ≈ 70% AA). In contrast, *T. ni* lacks germline dosage compensation: in the ovary, Z-linked transcript abundance is half that of autosomal transcripts (Z ≈ 50% AA), whereas in testis, Z-linked and autosomal transcripts have equal abundance (ZZ ≈ AA). We conclude that *T. ni*, like *B. mori* (*Walters and Hardcastle, 2011*), *Cydia pomonella* (*Gu et al., 2017*), and *Heliconius* butterflies (*Walters et al., 2015*), compensates for Z chromosome dosage in the soma by reducing gene expression in males, but does not decrease Z-linked gene expression in germline tissues.

Little is known about lepidopteran W chromosomes. The W chromosome is not included in the genome assembly of *Manduca sexta* (*Kanost et al., 2016*) or *B. mori* (*International Silkworm Genome Consortium, 2008*), and earlier efforts to assemble the silkworm W resulted in fragmented sequences containing transposons (*Abe et al., 2005*, *2008*; *Kawaoka et al., 2011*). The monarch genome scaffold continuity (N50 = 0.207 Mb versus N50 = 14.2 Mb for *T. ni*; (*Zhan et al., 2011*) is insufficient to permit assembly of a W chromosome. Our genome assembly includes the 2.92 Mb *T. ni* W chromosome comprising 32 contigs (contig N50 = 101 kb). In *T. ni*, W-linked contigs have higher repeat content, lower gene density, and lower transcriptional activity than autosomal or

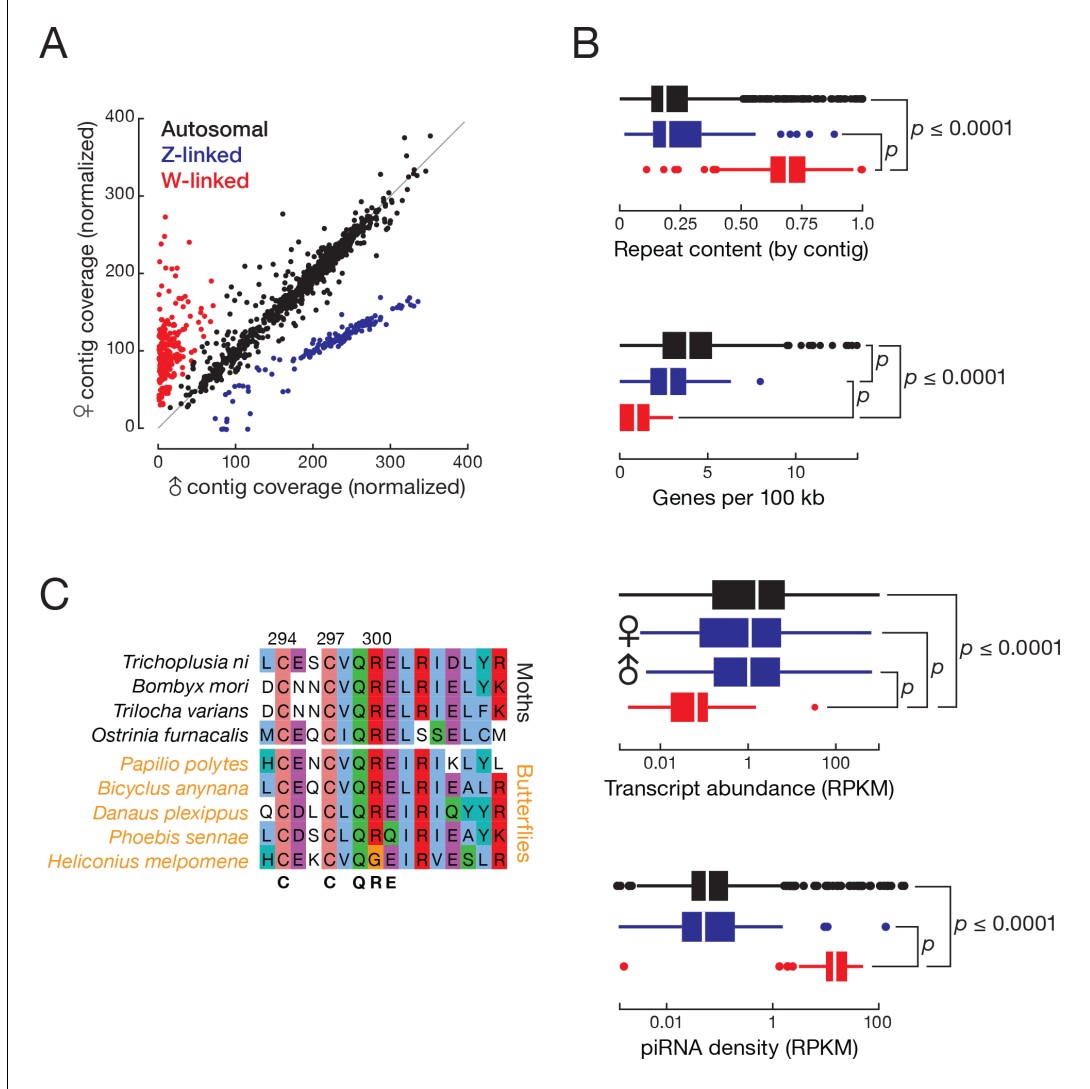

**Figure 2.** *T. ni* males are ZZ and females are ZW. (A) Normalized contig coverage in males and females. (B) Relative repeat content, gene density, transcript abundance (female and male thoraces), and piRNA density of autosomal, Z-linked, and W-linked contigs (ovary). (C) Multiple sequence alignment of the conserved region of the sex-determining gene *masc* among the lepidopteran species.

DOI: https://doi.org/10.7554/eLife.31628.008

The following figure supplements are available for figure 2:

**Figure supplement 1.** *T. ni* sex determination and dosage compensation.
DOI: https://doi.org/10.7554/eLife.31628.009
**Figure supplement 2.** CpG ratios and transposons.
DOI: https://doi.org/10.7554/eLife.31628.010

Z-linked contigs (*Figure 2B*). Other lepidopteran W chromosomes are similarly enriched in repeats and depleted of genes (*Abe et al., 2005*; *Fuková et al., 2005*; *Traut et al., 2007*).

A search for *T. ni* genes that are homologous to insect sex determination pathway genes detected *doublesex* (*dsx*), *masculinizer* (*masc*), *vitellogenin*, *transformer 2*, *intersex*, *sex lethal*, *ovarian tumor*, *ovo*, and *sans fille*. *T. ni* males produce a four-exon isoform of *dsx*, while females generate a six-exon *dsx* isoform (*Figure 2—figure supplement 1D*). The Lepidoptera-specific gene *masc* encodes a CCCH zinc finger protein. *masc* is associated with the expression of the sex-specific isoforms of *dsx* in lepidopterans, including silkworm (*Katsuma et al., 2015*). As in *B. mori*, *T. ni masc* lies next to the *scap* gene, supporting our annotation of *T. ni masc*. Lepidopteran *masc* genes are rapidly diverging and have low-sequence identity with one another (30.1%). *Figure 2C* shows the multiple

sequence alignment of the CCCH zinc finger domain of Masc proteins from several lepidopteran species.

## Telomeres and centromeres

Like many non-dipteran insects, *T. ni* has a single telomerase gene and telomeres containing TTAGG repeats (*Sahara et al., 1999*). We found 40 $(TTAGG)_n$ stretches longer than 100 nt (mean ± S.D. =600 ± 800 nt), nine at and 31 near contig boundaries (*Supplementary file 1F*; distance between $(TTAGG)_n$ and contig boundary = 5000 ± 6000 nt for the 40 stretches), indicating that our assembly captures the sequences of many telomeres. More than half (59%) of the sequences flanking the $(TTAGG)_n$ repeats are transposons, and ~49% of these belong to the non-long-terminal-repeat LINE/R1 family (*Supplementary file 1G*). These telomeric and subtelomeric characteristics of *T. ni* resemble those of *B. mori* (*Fujiwara et al., 2005*).

Lepidopteran chromosomes generally lack a coherent, monocentric centromere and are instead holocentric or diffuse (*Labbé et al., 2011*), and the silkworm, monarch butterfly, and diamondback moth genomes do not encode CenH3, a protein associated with monocentric chromosomes. The *T. ni* genome similarly does not contain a gene for CenH3, suggesting that its chromosomes are also holocentric.

## CpG content and DNA methylation

The *T. ni* genome is 35.6% GC, slightly less than *B. mori* (37.3%). The distributions of observed/expected CpG ratios in genes and across the genome (*Figure 2—figure supplement 2A*) reveal that *T. ni* is similar to other lepidopterans (silkworm, monarch butterfly, diamondback moth) and a coleopteran species (red flour beetle, *T. castaneum*), but different from honeybee and fruit fly. The honeybee genome has a high CpG content in genes and exhibits a bimodal CpG distribution across the genome as a whole; the fruit fly genome is uniformly depleted of CpG dinucleotides. The differences in CpG patterns reflect the presence of both the DNMT1 and DNMT3 DNA methyltransferases in the honeybee, the absence of either in fruit fly, and the presence of only DNMT1 in *T. ni*, *B. mori*, *D. plexippus*, *P. xylostella*, and *T. castaneum*. Thus, like many other insects, the *T. ni* genome likely has low levels of DNA methylation (*Xiang et al., 2010*; *Glastad et al., 2011*).

## Transposons and repeats

The *T. ni* genome contains 75.3 Mb of identifiable repeat elements (20.5% of the assembly), covering 458 repeat families (*Figure 2—figure supplement 2B*, *Supplementary file 1H*). With this level of repeat content, *T. ni* fits well with the positive correlation between genome size and repeat content among lepidopteran genomes (*Figure 2—figure supplement 2C*).

The DNA transposon piggyBac was originally isolated from a *T. ni* cell line (*Fraser et al., 1983*) and transposes effectively in a variety of species (*Lobo et al., 1999*; *Bonin and Mann, 2004*; *Wang et al., 2008*). We identified 262 copies of piggyBac in the Hi5 cell genome assembly. The family divergence rate of piggyBac is ~0.17%, substantially lower than other transposon families in the genome (*Supplementary file 1I* provides divergence rates for all transposon families). Among the individual piggyBac elements in the *T. ni* genome, 71 are specific to Hi5 cells. Compared to the 191 piggyBac insertions shared between *T. ni* and Hi5 cells (divergence rate = 0.22%), the Hi5-cell-specific elements are more highly conserved (divergence rate = 0.04%). We conclude that the piggyBac transposon entered the *T. ni* genome more recently than other transposons and, likely driven by the presence of many active piggyBac elements, expanded further during the immortalization of Hi5 cells in culture.

## miRNAs

miRNAs are ~22 nt non-coding RNAs that regulate mRNA stability and translation (*He and Hannon, 2004*; *Gao et al., 2005*). In insects, miRNA targets function in metamorphosis, reproduction, diapause, and other pathways of insect physiology and development (*Lucas and Raikhel, 2013*). To characterize the *T. ni* miRNA pathway, we sequenced RNA and small RNA from ovary, testis, thorax, and Hi5 cells. Then, we manually identified miRNA biogenesis genes such as *dcr-1*, *pasha*, *drosha*, and *ago2* (*Supplementary file 2A*) and computationally predicted 295 miRNA genes (*Figure 3*,

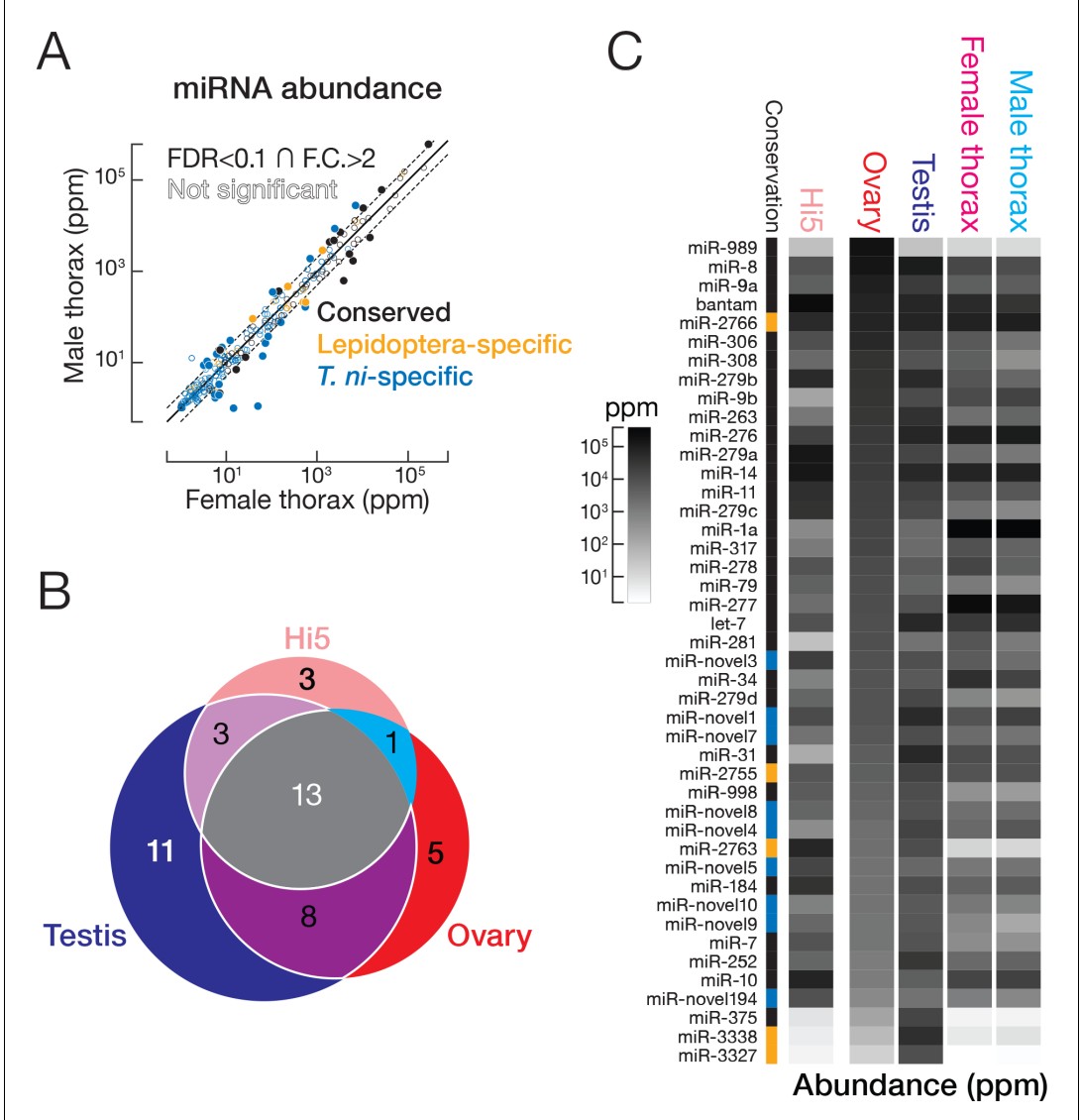

**Figure 3.** miRNA expression in *T. ni*. (**A**) Comparison of miRNA abundance in male and female *T. ni* thoraces. Solid circles, miRNAs with FDR < 0.1 and fold change >2. Outlined circles, all other miRNAs. (**B**) Comparison of the tissue distribution of the 44 most abundant miRNAs among *T. ni* ovaries, testes, and Hi5. (**C**) Heat map showing the abundance of miRNAs in (**B**). miRNAs are ordered according to abundance in ovary. Conservation status uses the same color scheme in (**A**).

DOI: https://doi.org/10.7554/eLife.31628.011

*Supplementary file 3A* and *Supplementary file 4*), including 77 conserved, 31 Lepidoptera-specific, and 187 novel, *T. ni*-specific miRNAs.

In thorax, 222 of 270 miRNAs had comparable abundance in males and females (≤2 fold difference or false discovery rate [FDR]≥0.1; *Figure 3A*). Of the 48 miRNAs having significantly different abundances in female and male thorax (>2 fold difference and FDR < 0.1; *Figure 3A*), miR-1a, let-7, and miR-278 were highly abundant (>1000 parts per million [ppm]) in either female or male thorax. miR-1a, a miRNA thought to be expressed in all animal muscle, was the most abundant miRNA in thorax in both sexes, but was 2.2-fold more abundant in males. miR-1 was previously shown to regulate muscle development in fruit flies (*Sokol and Ambros, 2005*) and to increase when locusts transition from solitary to swarming (*Wei et al., 2009*). *T. ni* let-7, which has the same mature miRNA sequence as its *D. melanogaster*, *C. elegans*, and mammalian counterparts (*Lagos-Quintana et al., 2001*) was also more abundant in males, whereas miR-278 was 2.6-fold more abundant in females.

let-7 may act in sex-specific pathways in metamorphosis (*Caygill and Johnston, 2008*), whereas miR-278 may play a sex-specific role in regulating energy homeostasis (*Teleman et al., 2006*).

A subset of less well-conserved miRNAs was also differentially expressed between male and female thorax. In general, poorly conserved miRNAs were less abundant: the median expression level for conserved miRNAs was 316 ppm, but only 161 ppm for Lepidoptera-specific and 4.22 ppm for *T. ni*-specific miRNAs. However, mir-2767, a Lepidoptera-specific miRNA, and three *T. ni*-specific miRNAs (mir-novel1, mir-novel4, mir-novel11) were both abundant (>1000 ppm) and differentially expressed in males and female thorax. We speculate that these recently evolved miRNAs may prove useful as targets for pest management.

Ovary, testis, and Hi5 cells have distinct miRNA expression profiles. We analyzed the expression patterns of the 44 most abundant miRNAs (*Figure 3B and C*), which explain 90% of miRNA reads in a tissue or cell line. Thirteen were expressed in ovaries, testes, and Hi5 cells. Of these 13, 11 were significantly more abundant in testis, 5 in ovary, and 3 in Hi5 cells (*Figure 3B*), suggesting that these miRNAs have important tissue- or cell-type-specific roles. miR-31 and miR-375, highly expressed in *T. ni* testis, are both mammalian tumor suppressors (*Creighton et al., 2010*; *Kinoshita et al., 2012*). miR-989, the most abundant miRNA in *T. ni* ovaries, plays an important role in border cell migration during *Drosophila* oogenesis (*Kugler et al., 2013*). miR-10, a miRNA in the Hox gene cluster, was preferentially expressed in Hi5 cells; its orthologs have been implicated in development and cancer (*Lund, 2010*), suggesting miR-10 played a role in the immortalization of the germline cells from which Hi5 cells derive.

## siRNAs

siRNAs, typically 20–22 nt long, regulate gene expression, defend against viral infection, and silence transposons (*Agrawal et al., 2003*; *van Rij et al., 2006*; *Sánchez-Vargas et al., 2009*; *Tyler et al., 2008*; *Tam et al., 2008*; *Zambon et al., 2006*; *Chung et al., 2008*; *Okamura et al., 2008b*; *Czech et al., 2008*; *Okamura et al., 2008b*; *Flynt et al., 2009*). They are processed by Dicer from double-stranded RNAs or hairpins into short double-stranded fragments bearing two-nucleotide, overhanging 3′ ends, which are subsequently loaded into Argonaute proteins (*Bernstein et al., 2001*; *Elbashir et al., 2001*; *Siomi and Siomi, 2009*). siRNAs require extensive sequence complementarity to their targets to elicit Argonaute-catalyzed target cleavage.

## Endogenous siRNAs from transposons and *cis*-NATs

Endogenous siRNAs (endo-siRNAs) can derive from transposon RNAs, *cis*-natural antisense transcripts (*cis*-NATs), and long hairpin RNAs (*Czech et al., 2008*; *Ghildiyal et al., 2008*; *Okamura et al., 2008a*; *Chung et al., 2008*; *Kawamura et al., 2008*; *Okamura et al., 2008a*; *Tam et al., 2008*; *Watanabe et al., 2008*) (hpRNAs). In *T. ni* ovary, testis, thorax, and Hi5 cells, 20.7–52.4% of siRNAs map to transposons, suggesting *T. ni* endogenous siRNAs suppress transposons in both the soma and the germline. Among the non-transposon siRNAs,<4.6% map to predicted hairpins, while 11.6–31.3% siRNAs map to *cis*-NATs (*Supplementary file 3B*).

## Exogenous siRNAs against a virus

Hi5 cells are latently infected with a positive-sense, bipartite alphanodavirus, TNCL virus (*Li et al., 2007*; *Andrew Ball and Johnson, 1998*) (Tn5 Cell Line virus). We asked if TNCL virus RNA is present in our *T. ni* samples and whether the RNAi pathway provides antiviral defense via TNCL virus-derived siRNAs. We detected no viral RNA in the *T. ni* ovary, testis, or thorax transcriptome, but both TNCL virus RNA1 (5010 fragments per kilobase of transcript per million mapped reads [FPKM]) and RNA2 (8280 FPKM) were readily found in the Hi5 transcriptome (*Figure 4A*). To test whether Hi5 cells mount an RNAi defense to TNCL virus infection, we mapped small RNA-seq reads that were not mappable to the *T. ni* genome to the two TNCL virus genomic segments. TNCL virus-mapping small RNAs showed a median length of 21 nt (modal length = 20 nt; *Figure 4A*), typical for siRNAs, suggesting that the Hi5 RNAi pathway actively combats the virus. The TNCL virus-mapping small RNAs bear the two-nucleotide, 3′ overhanging ends that are the hallmark of siRNAs (*Figure 4B*) (*Elbashir et al., 2001*). Moreover, the phased pattern of TNCL virus-mapping siRNAs suggests they are made one-after-another starting at the end of a dsRNA molecule: the distance between siRNA 5′ ends shows a periodicity of 20 nt, the length of a typical TNCL virus-mapping siRNA (*Figure 4C*). In

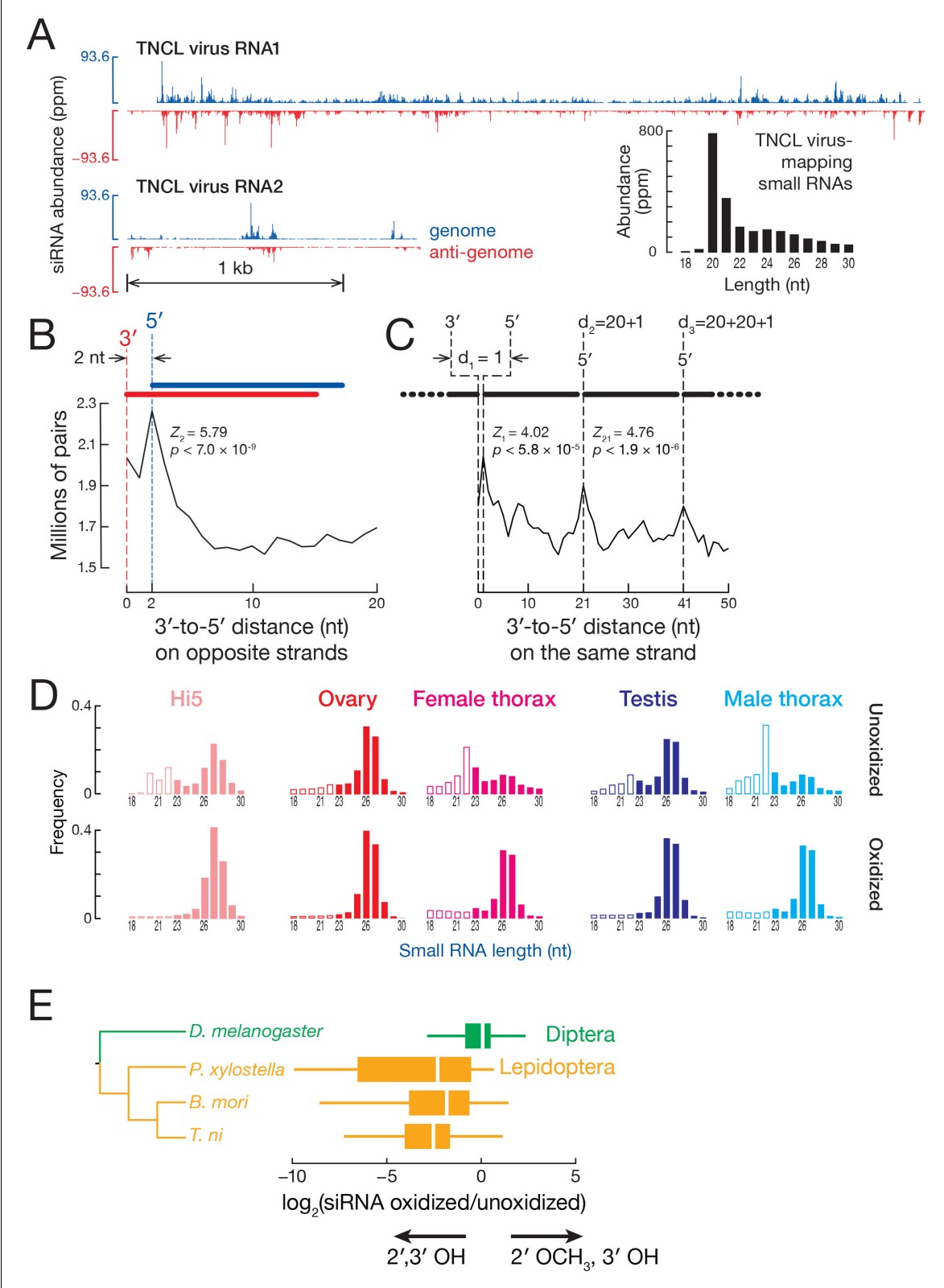

**Figure 4.** siRNA. (**A**) Distribution of siRNAs mapping to TNCL virus in the genomic (blue) and anti-genomic orientation (red). Inset: length distribution of TNCL virus-mapping small RNAs. (**B**) Distance between the 3′ and 5′ ends of siRNAs on opposite viral strands. (**C**) Distance between the 3′ and 5′ ends of siRNAs on the same viral strand. (**D**) Length distribution of small RNAs from unoxidized and oxidized small RNA-seq libraries. (**E**) Lepidopteran

*Figure 4 continued on next page*

*Figure 4 continued*

siRNAs are not 2′-*O*-methylated. The box plots display the ratio of abundance (as a fraction of all small RNAs sequenced) for each siRNA in oxidized versus unoxidized small RNA-seq libraries. The tree shows the phylogenetic relationships of the analyzed insects. Outliers are not shown.

DOI: https://doi.org/10.7554/eLife.31628.012

The following figure supplements are available for figure 4:

**Figure supplement 1.** *T. ni* siRNAs.

DOI: https://doi.org/10.7554/eLife.31628.013

**Figure supplement 2.** Loading asymmetry of siRNAs mapping to TNCL RNA1 (**A**) and RNA2 (**B**).

DOI: https://doi.org/10.7554/eLife.31628.014

*D. melanogaster*, Dicer-2 processively produces siRNAs, using ATP energy to translocate along a dsRNA molecule (*Cenik et al., 2011*). The phasing of anti-viral siRNAs in Hi5 cells suggests that *T. ni* Dicer-2 similarly generates multiple siRNAs from each molecule of dsRNA before dissociating.

In addition to siRNAs, the TNCL-mapping small RNAs include some 23–32 nt RNAs. These are unlikely to be anti-viral piRNAs, because they lack the characteristic first-nucleotide uridine bias and show no significant ping-pong signal (*Z*-score = −0.491). We conclude that Hi5 cells do not use piRNAs for viral defense.

## Lepidopteran siRNAs are not 2′-*O*-methylated

The discovery that the 3′ ends of *D. melanogaster* siRNAs, but not miRNAs, are 2′-*O*-methylated (*Pélisson et al., 2007*) led to the idea that insects in general methylate both siRNAs and piRNAs. Resistance to oxidation by $NaIO_4$ is the hallmark of 3′ terminal, 2′-*O*-methylation, and the enrichment of a small RNA in a high-throughput sequencing library prepared from $NaIO_4$-treated RNA suggests 2′-*O*-methylation. Conversely, depletion of small RNAs, such as miRNAs, from such an oxidized RNA library is strong evidence for unmodified 2′,3′ vicinal hydroxyl groups. Surprisingly, TNCL virus-mapping siRNAs were 130-fold depleted from our oxidized small RNA-seq library (22.0 ppm) compared to the unoxidized library (2870 ppm), suggesting that they are unmethylated. Sequencing of oxidized and unoxidized small RNA from *T. ni* ovary, testis, and thorax detected 20–22 nt peaks in unoxidized libraries; such peaks were absent from oxidized libraries (*Figure 4D*), suggesting that *T. ni* genome-mapping, endogenous siRNAs also lack 2′-O-methylation. We conclude that both *T. ni* exo- and endo-siRNAs are not 2′-*O*-methyl modified.

Are siRNAs unmethylated in other Lepidopteran species? We sequenced oxidized and unoxidized small RNAs from two additional Lepidoptera: *P. xylostella* and *B. mori*. Like *T. ni*, siRNAs from these Lepidoptera were abundant in libraries prepared from unoxidized small RNA but depleted from oxidized libraries (*Figure 4—figure supplement 1A*). The ratio of siRNAs in the oxidized library to siRNAs in the corresponding unoxidized library (ox/unox) provides a measure of siRNA 2′,3′ modification. For *D. melanogaster* siRNAs, the median ox/unox ratio was 1.00, whereas the three Lepidoptera species had median ox/unox ratios between 0.17 and 0.22 (*Figure 4E*), indicating their siRNAs were depleted from oxidized libraries and therefore bear unmodified 2′,3′ hydroxyl groups. We conclude that the last common ancestor of *T. ni*, *B. mori*, and *P. xylostella,* which diverged 170 mya, lacked the ability to 2′-*O*-methylate siRNA 3′ ends. We do not currently know whether the last common ancestor of Lepidoptera lost the capacity to methylate siRNAs or if some or all members of Diptera, the sister order of Lepidoptera, acquired this function, which is catalyzed by the piRNA-methylating enzyme Hen1 (*Saito et al., 2007*; *Horwich et al., 2007*; *Kirino and Mourelatos, 2007*).

Terminal 2′ methylation of *D. melanogaster* siRNAs is thought to protect them from non-templated nucleotide addition (tailing), 3′-to-5′ trimming, and wholesale degradation (*Ameres et al., 2010*). Since *T. ni* siRNAs lack a 2′-*O*-methyl group at their 3′ ends, we first asked if we could observe frequent trimming by examining shorter TNCL-mapping siRNA (18–19 nt). These siRNAs account for 1.05% of all TNCL-mapping siRNAs. They did not possess the typical siRNA one-after-another pattern ($Z_1$ = −0.674, p=0.500), yet more than 97.5% of these were prefixes of longer, phased siRNAs, indicating that these were trimmed siRNAs. We conclude that TNCL siRNA trimming is rare in Hi5 cells. We next asked whether *T. ni* and other lepidopteran siRNAs have higher frequencies of tailing. Despite the lack of 2′-*O*-methylation, most TNCL virus siRNAs were not tailed: just 6.69% of all virus-mapping small RNA reads contained 3′ non-templated nucleotides (*Figure 4—*

*figure supplement 1B*). Among the 3′ non-templated nucleotides, the most frequent addition was one or more uridines (49.6%) as observed previously for miRNAs and siRNAs in other animals (*Ameres et al., 2010*; *Chou et al., 2015*). Endogenous siRNA tailing frequencies for the lepidopterans *T. ni* (10.2%, ovary), *B. mori* (5.97%, eggs), and *P. xylostella* (8.58%, ovary) were also similar to *D. melanogaster* (6.71%, ovary). We speculate that lepidopterans have other mechanisms to maintain siRNA stability or that trimming and tailing in lepidopterans are less efficient than in flies.

siRNAs are non-randomly loaded into Argonaute proteins: the guide strand, the strand with the more weakly base paired 5′ end, is favored for loading (*Khvorova et al., 2003*; *Schwarz et al., 2003*); the disfavored passenger strand is destroyed. Thus, loading skews the abundance of the two siRNA strands. To test if non-methylated siRNAs are loaded into Argonaute, we computationally paired single-stranded siRNAs that compose an siRNA duplex bearing two-nucleotide overhanging 3′ ends and calculated the relative abundance of the two siRNA strands. For TNCL-mapping siRNAs, 72.3% of siRNA duplexes had guide/passenger strand ratios $\geq 2$ (median = 3.90; mean = 10.2; *Figure 4—figure supplement 2*). Among genome-mapping, 20–22 nt small RNAs 78.5% of duplexes had guide/passenger strand ratios $\geq 2$ (median 5.44; average 56.2). We conclude that the majority of exogenous and endogenous siRNAs are loaded, presumably into Ago2.

## piRNAs

In animals, piRNAs,~23–32 nt long, protect the germline genome by suppressing the transcription or accumulation of transposon and repetitive RNA (*Girard et al., 2006*; *Lau et al., 2006*; *Vagin et al., 2006*; *Brennecke et al., 2007*; *Aravin et al., 2007*). In *D. melanogaster*, dedicated transposon-rich loci (piRNA clusters) give rise to piRNA precursor transcripts, which are processed into piRNAs loaded into one of three PIWI proteins, Piwi, Aubergine (Aub), or Argonaute3 (Ago3). Piwi acts in the nucleus to direct tri-methylation of histone H3 on lysine nine on transposon and repetitive genomic sequences (*Sienski et al., 2012*; *Le Thomas et al., 2014a*, *2014b*). In fly cytoplasm, piRNAs guide the Piwi paralog Aub to cleave transposon mRNAs. The mRNA cleavage products can then produce more piRNAs, which are loaded into Ago3. In turn, these sense piRNAs direct Ago3 to cleave transcripts from piRNA clusters, generating additional piRNAs bound to Aub. The resulting 'Ping-Pong' feed-forward loop both amplifies piRNAs and represses transposon activity (*Brennecke et al., 2007*; *Gunawardane et al., 2007*). Finally, Ago3 cleavage not only produces Aub-bound piRNAs, but also initiates the production of Piwi-bound, phased piRNAs that diversify the piRNA pool (*Mohn et al., 2015*; *Han et al., 2015b*).

## piRNA pathway proteins

The *T. ni* genome contains a full repertoire of genes encoding piRNA pathway proteins (*Supplementary file 2B*). These genes were expressed in both germline and somatic tissues, but were higher in ovary, testis, and Hi5 cells compared to thorax (median ratios: ovary/thorax = 14.2, testis/thorax = 2.9, and Hi5/thorax = 4.9; *Figure 5A*). Expression of piRNA pathway genes in the Hi5 cell line suggests that it recapitulates the germline piRNA pathway. Although most *T. ni* piRNA pathway genes correspond directly to their *D. melanogaster* orthologs, *T. ni* encodes only two PIWI proteins, TnPiwi and TnAgo3. The fly proteins Aub and Piwi are paralogs that arose from a single ancestral PIWI protein after the divergence of flies and mosquitos (*Lewis et al., 2016*). We do not yet know whether TnPiwi functions more like *Drosophila* Aub or Piwi. In *D. melanogaster*, piRNA clusters—the genomic sources of most transposon-silencing germline piRNAs—are marked by the proteins Rhino, Cutoff, and Deadlock, which allow transcription of these heterochromatic loci (*Klattenhoff et al., 2009*; *Pane et al., 2011*; *Mohn et al., 2014*; *Zhang et al., 2014*). *T. ni* lacks detectable Rhino, Cutoff, and Deadlock orthologs. In fact, this trio of proteins is poorly conserved, and the mechanism by which they mark fly piRNA source loci may be unique to Drosophilids. In this regard, *T. ni* likely provides a more universal insect model for the mechanisms by which germ cells distinguish piRNA precursor RNAs from other protein-coding and non-coding transcripts.

## piRNA cluster architecture

In both the germline and the soma, *T. ni* piRNAs originate from discrete genomic loci. To define these piRNA source loci, we employed an expectation-maximization algorithm that resolves piRNAs mapping to multiple genomic locations. Applying this method to multiple small RNA-seq datasets,

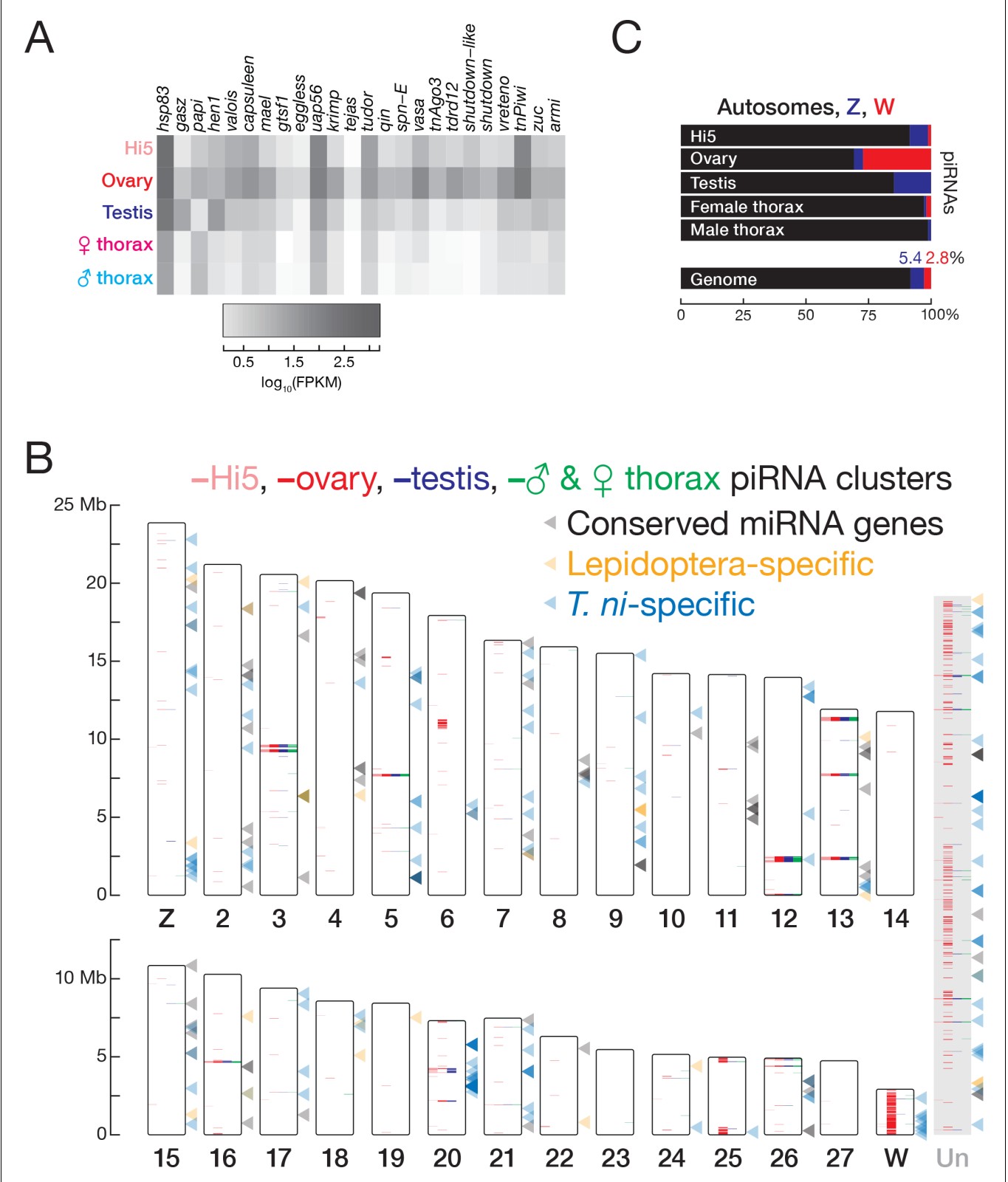

**Figure 5.** piRNAs and miRNAs in the *T. ni* genome. (**A**) Abundance of mRNAs encoding piRNA pathway proteins) in Hi5 cells, ovary, testis, and thorax. (**B**) Ideogram displaying the positions of miRNA genes (arrowheads) and piRNA clusters in the *T. ni* genome. Color-coding reports tissue expression for Hi5 cells, ovaries, testis, and thorax. Contigs that cannot be placed onto chromosome-length scaffolds are arbitrarily concatenated and are marked

*Figure 5 continued on next page*

*Figure 5 continued*

'Un.' (C) Distribution of piRNAs among the autosomes, Z, and W chromosomes in Hi5 cells, ovary, testis, and female and male thorax, compared with the fraction of the genome corresponding to autosomes, W, and Z chromosomes.

DOI: https://doi.org/10.7554/eLife.31628.015

The following figure supplements are available for figure 5:

**Figure supplement 1.** piRNA abundance (ppm) along the most productive piRNA cluster.

DOI: https://doi.org/10.7554/eLife.31628.016

**Figure supplement 2.** *T. ni* piRNAs.

DOI: https://doi.org/10.7554/eLife.31628.017

we defined piRNA-producing loci comprising 10.7 Mb (348 clusters) in ovary, 3.1 Mb (79 clusters) in testis, 3.0 Mb (71 clusters) in Hi5 cells, and 2.4 Mb (65 clusters) in thorax (*Figure 5B*). For each tissue or cell-type, these 393 clusters explain >70% of uniquely mapped piRNAs and >70% of all piRNAs when using expectation-maximization mapping. A core set of piRNA-producing loci comprising 1.5 Mb is active in both germline and somatic tissues.

*T. ni* piRNA clusters vary substantially in size and expression level. In ovary, half the bases in piRNA clusters are in just 67 loci, with a median length of 53 kb. Among these, five span >200 kb, while the smallest is just 38 kb. The most productive piRNA source is a 264 kb locus on chromosome 13 (*Figure 5—figure supplement 1*); 7.8% of uniquely mapped piRNAs—50,000 distinct piRNA sequences—reside in this locus. Collectively, the top 20 ovary piRNA loci explain half the uniquely mapped piRNAs, yet constitute only 0.7% of the genome. Globally, 61.9% of bases in piRNA clusters are repetitive, and 74.5% transposon-mapping piRNAs are antisense, suggesting that *T. ni* uses anti-sense piRNAs to suppress transposon transcripts.

In the fly ovary germline, most piRNA clusters generate precursor RNAs from both DNA strands. These dual-strand clusters fuel the 'Ping-Pong' amplification cycle (*Brennecke et al., 2007*; *Gunawardane et al., 2007*). Other fly piRNA clusters, such as the paradigmatic *flamenco* gene (*Prud'homme et al., 1995*; *Brennecke et al., 2007*; *Pélisson et al., 2007*; *Malone et al., 2009*; *Goriaux et al., 2014*) are transcribed from one strand only and are organized to generate antisense piRNAs directly, without further Ping-Pong amplification (*Malone et al., 2009*). These uni-strand clusters are the only sources of piRNAs in the follicle cells, somatic cells that support fly oocyte development and express only a single PIWI protein, Piwi (*Malone et al., 2009*).

The *T. ni* genome contains both dual- and uni-strand piRNA clusters. In ovary, 62 of 348 piRNA-producing loci are dual-strand (Watson/Crick > 0.5 or Watson/Crick < 2). These loci produce 35.9% of uniquely mapped piRNAs and 22.8% of all piRNAs; 71.6% of transposon-mapping piRNA reads from these loci are antisense. The remaining 286 uni-strand loci account for 54.8% of uniquely mapped piRNAs and 36.7% of all piRNAs. Most piRNAs (74.8% of reads) from uni-strand clusters are antisense to transposons, the orientation required for repressing transposon mRNA accumulation. At least part of the piRNA antisense bias reflects positive selection for antisense insertions in uni-strand clusters: 57.1% of transposon insertions—79.7% of transposon-mapping nucleotides—are opposite the direction of piRNA precursor transcription, significantly different from dual-strand clusters, in which transposons are inserted randomly: 49.5% of transposon insertions in dual-strand clusters are in the antisense direction (*Figure 5—figure supplement 2A*). For one 77 kb uni-strand cluster on chromosome 20, 99.0% of piRNA reads (96% of piRNA sequences) that can be uniquely assigned are from the Crick strand, while 67.6% of transposon insertions and 79.7% of transposon-mapping nucleotides at this locus lie on the Watson strand.

## Nearly the entire W chromosome produces piRNAs

The largest ovary cluster is a 462 kb W-linked region, consistent with our finding that the W chromosome is a major source of piRNAs (*Figure 5B and C* and *Figure 5—figure supplement 2B*). Our data likely underestimates the length of this large piRNA cluster, as it is difficult to resolve reads mapping to its flanking regions: 70.8% of bases in the flanking regions do not permit piRNAs to map uniquely to the genome. In fact, 85.1% of the sequences between clusters on the W chromosome are not uniquely mappable. These gaps appear to reflect low mappability and not boundaries between discrete clusters. We propose that the W chromosome itself is a giant piRNA cluster.

To further test this idea, we identified piRNA reads that uniquely map to one location among all contigs and measured their abundance per kilobase of the genome. W-linked contigs had a median piRNA abundance of 14.4 RPKM in ovaries, 379-fold higher than the median of all autosomal and Z-linked contigs, consistent with the view that almost the entire W chromosome produces piRNAs. In *B. mori* females, a plurality of piRNAs come from the W chromosome: ovary-enriched piRNAs often map to W-linked sequences, but not autosomes (*Kawaoka et al., 2011*). Similarly, for *T. ni*, 27.2% of uniquely mapping ovary piRNAs derive from W-linked sequences, even though these contigs compose only 2.8% of the genome (*Figure 5C*). The W chromosome may produce more piRNAs than our estimate, as the unassembled repetitive portions of the W chromosome likely also produce piRNAs. Thus, the entire W chromosome is a major source of piRNAs in *T. ni* ovaries (*Figure 5B*). To our knowledge, the *T. ni* W chromosome is the first example of an entire chromosome devoted to piRNA production.

To determine if there are W-linked regions devoid of piRNAs, we mapped all piRNAs to the W-linked contigs and found that 11.0% of the W-linked bases were not covered by any piRNAs, indicating at least part of the W chromosome does not produce any piRNAs. Next, we manually inspected 74 putative W-linked protein-coding genes and nine putative W-linked miRNAs. All nine W-linked miRNAs (*Figure 5B*, *Supplementary file 1J*) are *T. ni*-specific, and small RNAs mapping to these predicted miRNA loci showed significant ping-pong signature (Z-score = 14.2, p=1.81 $\times$ 10$^{-45}$), suggesting that these are likely piRNAs, not authentic miRNAs. For the putative protein-coding genes, we categorized them into orphan genes (no homologs found), transposons (good homology to transposons), uncharacterized/hypothetical proteins, and potential protein-coding genes with homology to the NCBI non-redundant protein sequences. We then asked whether piRNAs were produced from these genes (*Figure 5—figure supplement 2C*). Among W-linked genes, those with transposon homology on average produced the most piRNAs (44.9 median ppm), whereas those with homology to annotated genes produced the fewest (9.81 median ppm). Some putative genes (such as TNI001015 and TNI005339) produced no piRNAs at all. We conclude that although some W-linked loci do not produce piRNAs, nearly the entire W chromosome produces piRNAs.

In contrast to the W chromosome, *T. ni* autosomes and the Z chromosome produce piRNAs from discrete loci—63 autosomal and 11 Z-linked contigs had piRNA levels > 10 rpkm. Few piRNAs are produced outside of these loci: for example, the median piRNA level across all autosomal and Z-linked contigs was ~0 in ovaries (*Figure 5—figure supplement 2B*).

## Expression of piRNA clusters

In the *T. ni* germline, piRNA production from individual clusters varies widely, but the same five piRNA clusters produce the most piRNAs in ovary (34.9% of piRNAs), testis (49.3%), and Hi5 cells (44.0%), suggesting that they serve as master loci for germline transposon silencing. Other piRNA clusters show tissue-specific expression, with the W chromosome producing more piRNAs in ovary than in Hi5 cells, and three Z-linked clusters producing many more piRNAs in testis than in ovary (15.0–24.7 times more), even after accounting for the absence of dosage compensation in germline tissues (*Figure 6—figure supplement 1A*).

Hi5 cells are female, yet many piRNA-producing regions of the W chromosome that are active in the ovary produce few piRNAs in Hi5 cells (*Figure 6—figure supplement 1A*). We do not know whether this reflects a reorganization of cluster expression upon Hi5 cell immortalization or if Hi5 cells correspond to a specific germ cell type that is underrepresented in whole ovaries. At least 40 loci produce piRNAs in Hi5 cells but not in ovaries. Comparison of DNA-seq data from *T. ni* and Hi5 identified 74 transposon insertions in 12 of the Hi5-specific piRNA clusters. Older transposons have more time to undergo sequence drift from the consensus sequence of the corresponding transposon family. The 74 Hi5-specific transposon insertions, which include both DNA and LTR transposons, had significantly lower divergence rates than those common to ovary and Hi5 cells (*Figure 6A*), consistent with the idea that recent transposition events generated the novel piRNA clusters in Hi5 cells. We conclude that the Hi5-specific piRNA-producing loci are quite young, suggesting that *T. ni* and perhaps other lepidopterans can readily generate novel piRNA clusters.

piRNA clusters active in thorax occupy ~0.57% of the genome and explain 86.8% of uniquely mapped somatic piRNAs in females and 89.5% in males. More than 90% of bases in clusters expressed in thorax are shared with clusters expressed in ovary (*Figure 6—figure supplement 1B*).

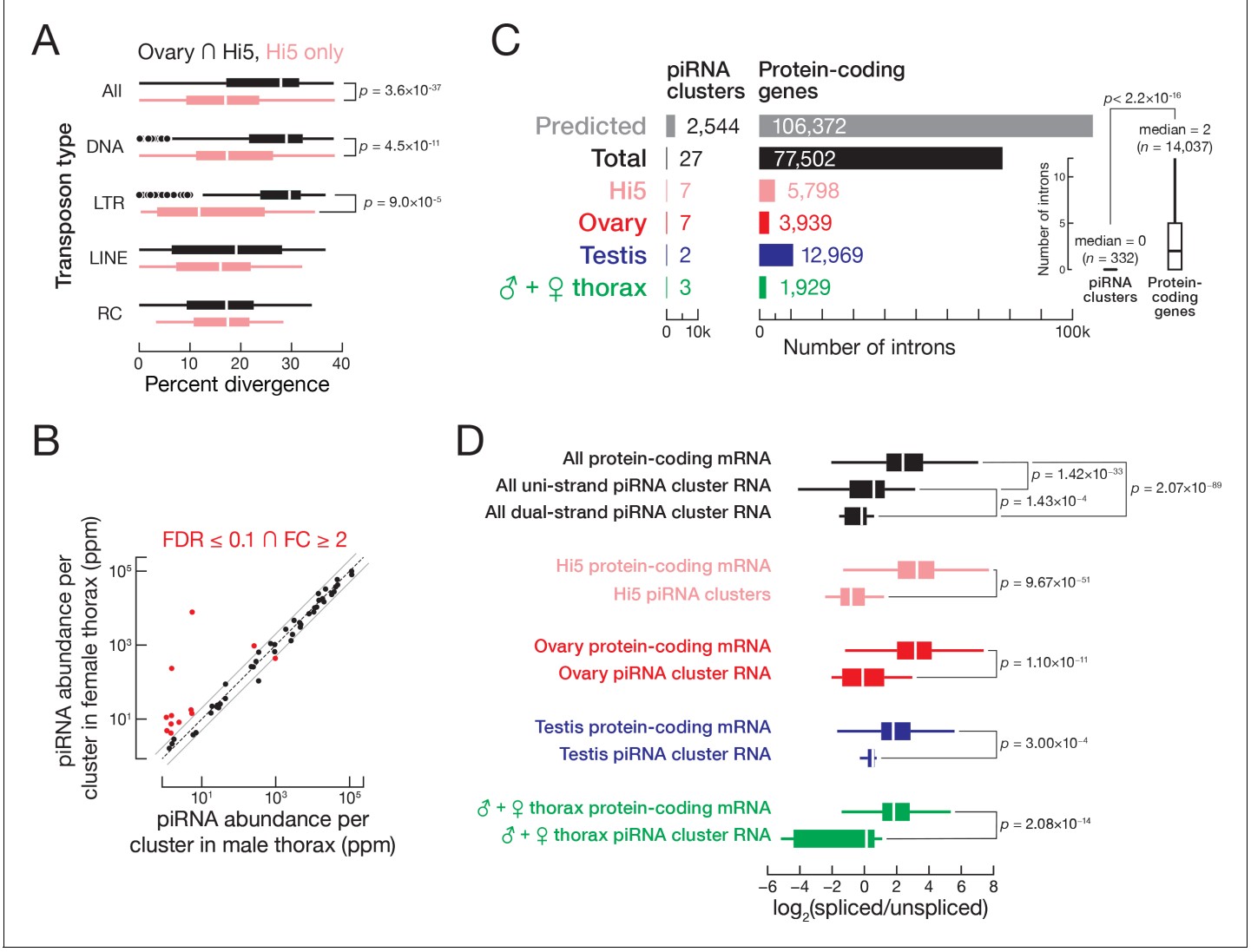

**Figure 6.** *T. ni* piRNAs. (**A**) Hi5-specific piRNA clusters contain younger transposon copies. RC, rolling-circle transposons; LINE, Long interspersed nuclear elements; LTR, long terminal repeat retrotransposon; DNA, DNA transposon. (**B**) Comparison of piRNA abundance per cluster in female and male thorax. (**C**) piRNA precursors are rarely spliced. The number of introns supported by exon-exon junction-mapping reads is shown for protein-coding genes and for piRNA clusters for each tissue or cell type. (**D**) piRNA precursors are inefficiently spliced. Splicing efficiency is defined as the ratio of spliced over unspliced reads. Splice sites were categorized into those inside and outside piRNA clusters. Outliers are not shown.

DOI: https://doi.org/10.7554/eLife.31628.018

The following figure supplement is available for figure 6:

**Figure supplement 1.** *T. ni* piRNA clusters.

DOI: https://doi.org/10.7554/eLife.31628.019

Such broadly expressed clusters explain 83.7% of uniquely mapping piRNAs in female thorax and 86.1% in male thorax. Thus, the majority of piRNAs in the *T. ni* soma come from clusters that are also active in the germline. In general, autosomal piRNA cluster expression is similar between female and male thorax, but 12 clusters are differentially expressed between male and female thorax. Of these, nine are W-linked clusters that produce significantly more piRNAs in female than in male thorax (*Figure 6B*).

## piRNA precursor transcripts are rarely spliced

In *D. melanogaster*, Rhino suppresses splicing of piRNA precursors transcribed from dual-strand piRNA clusters (*Mohn et al., 2014*; *Zhang et al., 2014*). Fly uni-strand piRNA clusters do not bind

Rhino and behave like canonical RNA polymerase II transcribed genes (*Brennecke et al., 2007*; *Goriaux et al., 2014*). Although *T. ni* has no *rhino* ortholog, its piRNA precursor RNAs are rarely spliced as observed for clusters in flies. We identified splicing events in our RNA-seq data, requiring $\geq 10$ reads that map across exon-exon junctions and a minimum splicing entropy of 2 to exclude PCR duplicates (*Graveley et al., 2011*). This approach detected just 27 splice sites among all piRNA precursor transcripts from ovary, testis, thorax, and Hi5 piRNA clusters (*Figure 6C*). Of these 27 splice sites, 19 fall in uni-strand piRNA clusters. We conclude that, as in flies, transcripts from *T. ni* dual-strand piRNAs clusters are rarely if ever spliced. Unlike flies (*Goriaux et al., 2014*), RNA from *T. ni* uni-strand piRNA clusters also undergoes splicing infrequently.

The absence of piRNA precursor splicing in dual-strand piRNA clusters could reflect an active suppression of the splicing machinery or a lack of splice sites. To distinguish between these two mechanisms, we predicted gene models for piRNA-producing loci, employing the same parameters used for protein-coding genes. For piRNA clusters, this approach generated 1332 gene models encoding polypeptides > 200 amino acids. These models comprise 2544 introns with consensus splicing signals (*Figure 6—figure supplement 1C*). Notably, ~90% of these predicted gene models had high sequence similarity to transposon consensus sequences (BLAST e-value <$10^{-10}$), indicating that many transposons in piRNA clusters have intact splice sites. We conclude that piRNA precursors contain splice sites, but their use is actively suppressed.

To measure splicing efficiency, we calculated the ratio of spliced to unspliced reads for each predicted splice site in the piRNA clusters. High-confidence splice sites in protein-coding genes outside piRNA clusters served as a control. Compared to the control set of genes, splicing efficiency in piRNA loci was 9.67-fold lower in ovary, 2.41-fold lower in testis, 3.23-fold lower in thorax, and 17.0-fold lower in Hi5 cells (*Figure 6D*), showing that *T. ni* piRNA precursor transcripts are rarely and inefficiently spliced. To test whether uni- and dual-strand piRNA cluster transcripts are differentially spliced in *T. ni*, we evaluated the experimentally supported splice sites from Hi5, ovary, testis, and thorax collectively. Dual-strand cluster transcripts had 1.71-fold lower splicing efficiency compared to uni-strand clusters (*Figure 6D*). Thus, *T. ni* suppresses splicing of dual- and uni-strand piRNA cluster transcripts by a mechanism distinct from the Rhino-dependent pathway in *D. melanogaster*. That this novel splicing suppression pathway is active in Hi5 cells should facilitate its molecular dissection.

## Genome-editing and single-cell cloning of Hi5 cells

The study of arthropod piRNAs has been limited both by a lack of suitable cultured cell models and by the dominance of *D. melanogaster* as a piRNA model for arthropods generally. Although Vasa-positive *D. melanogaster* ovarian cells have been isolated and cultured (*Niki et al., 2006*), no dipteran germ cell line is currently available. *D. melanogaster* somatic OSS, OSC and Kc167 cells produce piRNAs, but lack key features of the canonical germline pathway (*Lau et al., 2009*; *Saito et al., 2009*; *Vrettos et al., 2017*). In addition to Hi5 cells, lepidopteran cell lines from *Spodoptera frugiperda* (Sf9) and *B. mori* (BmN4) produce germline piRNAs (*Kawaoka et al., 2009*). The *S. frugiperda* genome remains a draft with 37,243 scaffolds and an N50 of 53.7 kb (*Kakumani et al., 2014*). Currently, the BmN4 cell line is the only ex vivo model for invertebrate germline piRNA biogenesis and function. The *B. mori* genome sequence currently comprises 43,463 scaffolds with an N50 of 4.01 Mb (*International Silkworm Genome Consortium, 2008*). Unfortunately, BmN4 cells readily differentiate into two morphologically distinct cell types (*Iwanaga et al., 2014*). Although genome editing with Cas9 has been demonstrated in BmN4 cells (*Zhu et al., 2015*), no protocols for cloning individual, genome-modified BmN4 cells have been reported (*Mon et al., 2004*; *Kawaoka et al., 2009*; *Honda et al., 2013*). In contrast, Hi5 cells are cultured using commercially available media, readily transfected, and, we report here, efficiently engineered with Cas9 and grown from single cells into clonal lines.

The bacterial DNA nuclease Cas9, targeted by a single guide RNA (sgRNA), enables rapid and efficient genome editing in worms, flies, and mice, as well as in a variety of cultured animal cell lines (*Jinek et al., 2012*; *Barrangou and Horvath, 2017*; *Komor et al., 2017*). The site-specific double-strand DNA breaks catalyzed by Cas9 can be repaired by error-prone non-homologous end joining (NHEJ), disrupting a protein-coding sequence or, when two sgRNAs are used, deleting a region of genomic DNA. Alternatively, homology-directed repair (HDR) using an exogenous DNA template allows the introduction of novel sequences, including fluorescent proteins or epitope tags, as well as point mutations in individual genes (*Cong et al., 2013*).

As a proof-of-concept, we used Cas9 and two sgRNAs to generate a deletion in the piRNA pathway gene *TnPiwi*. The two sgRNAs, whose target sites lie 881 bp apart (**Figure 7A**), were transcribed in vitro, loaded into purified, recombinant Cas9 protein, and the resulting sgRNA/Cas9 ribonucleoprotein complexes (RNPs) transfected into Hi5 cells. PCR of genomic DNA isolated 48 hr later was used to detect alterations in the *TnPiwi* gene. A novel PCR product, ~900 bp smaller than the

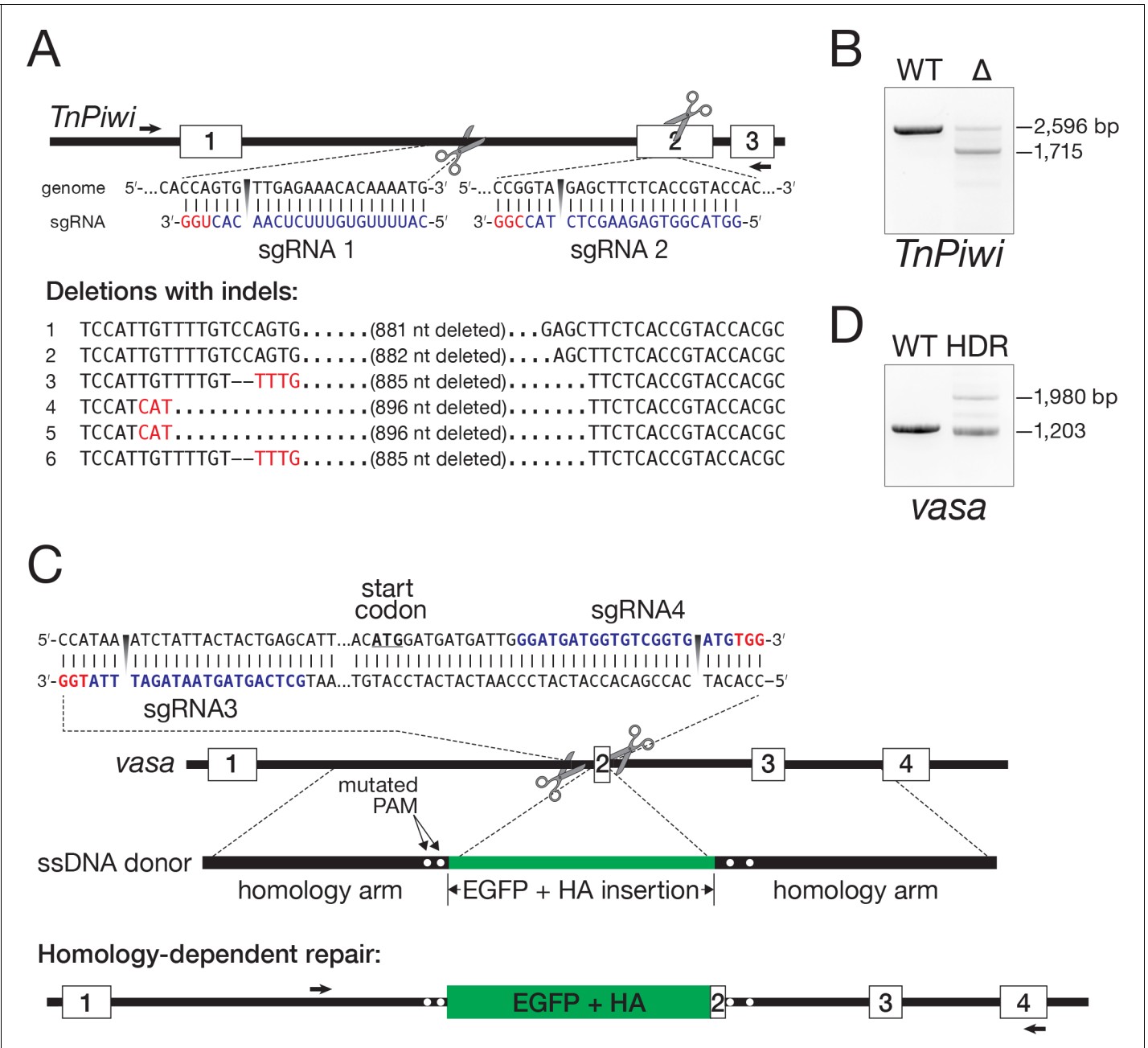

**Figure 7.** Genome editing in Hi5 cells. (**A**) Strategy for using Cas9/sgRNA RNPs to generate a loss-of-function *TnPiwi* deletion allele. Red, protospacer-adjacent motif (PAM); blue, protospacer sequence. Arrows indicate the diagnostic forward and reverse primers used in PCR to detect genomic deletions (Δ). Sanger sequencing of the ~1700 bp PCR products validated the *TnPiwi* deletions. (**B**) An example of PCR analysis of a *TnPiwi* deletion event. (**C**) Strategy for using Cas9/sgRNA RNPs and a single-stranded DNA homology donor to insert EGFP and an HA-tag in-frame with the *vasa* open-reading frame. (**D**) An example of PCR analysis of a successful HDR event. DNA isolated from wild type (WT) and FACS-sorted, EGFP-expressing Hi5 cells (HDR) were used as templates.

DOI: https://doi.org/10.7554/eLife.31628.020

product amplified using DNA from control cells, indicated that the desired deletion had been created (*Figure 7B*). Sanger sequencing of the PCR products confirmed deletion of 881–896 bp from the *TnPiwi* gene. The presence of indels—short deletions and non-templated nucleotide additions—at the deletion junction is consistent with a Cas9-mediated dsDNA break having been repaired by NHEJ (*Figure 7A*). We note that these cells still contain at least one wild-type copy of *TnPiwi*. We have not yet obtained cells in which all four copies of *TnPiwi* are disrupted, perhaps because in the absence of Piwi, Hi5 cells are inviable.

To test whether an exogenous donor DNA could facilitate the site-specific incorporation of protein tag sequences into Hi5 genome, we designed two sgRNAs with target sites ~ 90 bp apart, flanking the *vasa* start codon (*Figure 7C*). As a donor, we used a single-stranded DNA (ssDNA) encoding EGFP and an HA epitope tag flanked by genomic sequences 787 bp upstream and 768 bp downstream of the *vasa* start codon (*Figure 7C*). Cas9 and the two sgRNAs were cotransfected with the ssDNA donor, and, 1 week later, EGFP-positive cells were detected by fluorescence microscopy. PCR amplification of the targeted region using genomic DNA from EGFP-expressing cells confirmed integration of EGFP and the HA tag into the *vasa* gene (*Figure 7D*). Sanger sequencing further confirmed integration of EGFP and the HA tag in-frame with the *vasa* open-reading frame (*Supplemental file 9*).

To establish a clonal line from the EGFP-HA-tagged Vasa-expressing cells, individual EGFP-positive cells were isolated by FACS and cultured on selectively permeable filters above a feeder layer of wild-type Hi5 cells (*Figure 8A*). Growth of the genome-modified single cells required live Hi5 feeder cells—conditioned media did not suffice—presumably because the feeder cells provide short-lived growth factors or other trophic molecules. Single EGFP-positive clones developed 1 month after seeding and could be further grown without feeder cells as a clonally derived cell line (*Figure 8B*).

## Hi5 cell Vasa is present in a nuage-like, perinuclear structure

In the germline of *D. melanogaster* and other species, components of the piRNA biogenesis pathway, including Vasa, Aub, Ago3, and multiple Tudor-domain proteins, localize to a perinuclear structure called nuage (*Eddy, 1975*; *Findley et al., 2003*; *Lim and Kai, 2007*; *Li et al., 2009*; *Liu et al., 2011a*; *Webster et al., 2015*). Vasa, a germline-specific nuage component, is widely used as a marker for nuage. In BmN4 cells, transiently transfected Vasa localizes to a perinuclear structure resembling nuage (*Xiol et al., 2012*; *Patil et al., 2017*). To determine whether nuage-like structures are present in Hi5 cells, we examined Vasa localization in the Hi5 cells in which the endogenous *vasa* gene was engineered to fuse EGFP and an HA epitope tag to the Vasa amino-terminus. We used two different immunostaining strategies to detect the EGFP-HA-Vasa fusion protein: a mouse monoclonal anti-GFP antibody and a rabbit monoclonal anti-HA antibody. GFP and HA colocalized in a perinuclear structure, consistent with Vasa localizing to nuage in Hi5 cells (*Figure 8C*).

## Discussion

Using Hi5 cells, we have sequenced and assembled the genome of the cabbage looper, *T. ni*, a common and destructive agricultural pest that feeds on many plants of economic importance. Examination of the *T. ni* genome and transcriptome reveals the expansion of detoxification-related gene families (*Table 1* and *Supplementary file 6*), many members of which are implicated in insecticide resistance and are potential targets of pest control. The *T. ni* genome should enable study of the genetic diversity and population structure of this generalist pest, which adapts to different environmental niches worldwide. Moreover, as the sister order of Diptera, Lepidoptera like *T. ni* provide a counterpoint for the well-studied insect model *D. melanogaster*.

The use of Hi-C sequencing was an essential step in assembling the final 368.2 Mb *T. ni* genome into high-quality, chromosome-length scaffolds. The integration of long reads, short reads, and Hi-C provides a rapid and efficient paradigm for generating chromosome-level assemblies of other animal genomes. This strategy assembled the gene-poor, repeat-rich *T. ni* W chromosome, which is, to our knowledge, the first chromosome-level sequence of a lepidopteran W chromosome. Our analysis of autosomal, Z-linked, and W-linked transcripts provides insights into lepidopteran dosage compensation and sex determination. Our data show that *T. ni* compensates for Z chromosome dosage in the soma by reducing transcription of both Z homologs in males, but Z dosage is uncompensated in the germline.

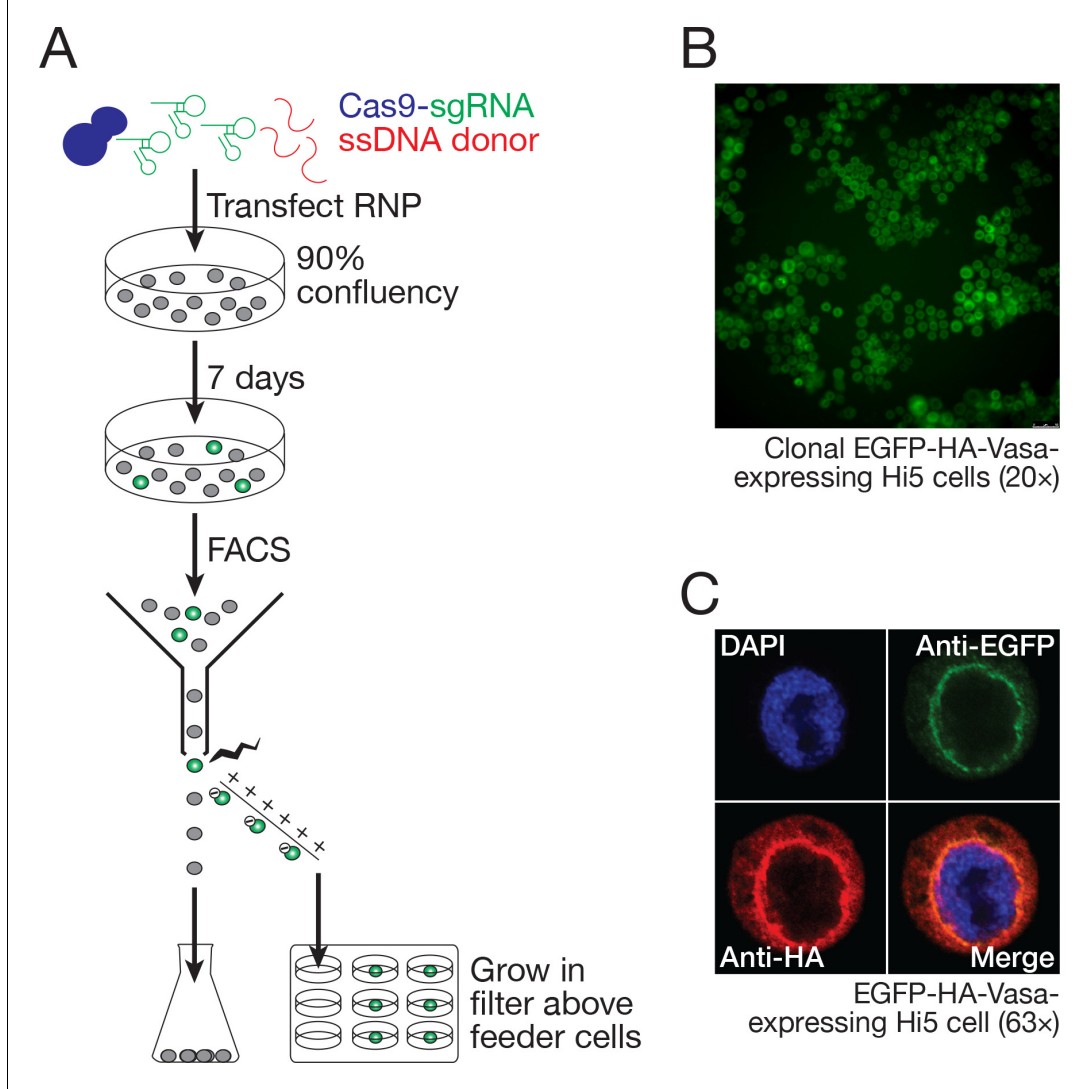

**Figure 8.** Hi5 cells contain nuage. (**A**) Schematic of single-clone selection of genome-edited Hi5 cells using the strategy described in *Figure 7C*. (**B**) A representative field of Hi5 cells edited to express EGFP-HA-Vasa from the endogenous locus. (**C**) A representative image of a fixed, EGFP-HA-Vasa-expressing Hi5 cell stained with DAPI, anti-EGFP and anti-HA antibodies. EGFP and HA staining colocalize in a perinuclear structure consistent with Vasa localizing to nuage.

DOI: https://doi.org/10.7554/eLife.31628.021

In addition to long RNAs, we characterized miRNAs, siRNAs, and piRNAs in *T. ni* gonads, soma, and cultured Hi5 cells. miRNAs are widely expressed in *T. ni* tissues, providing examples of germ-line-enriched and somatic miRNAs, as well as highly conserved, lepidopteran-specific, and novel *T. ni* miRNAs. Like flies, *T. ni* possess siRNAs that map to transposons, *cis*-NATs and hpRNAs. Unexpectedly, *T. ni* siRNAs—and likely all lepidopteran siRNAs—lack a 2′-*O*-methyl modification at their 3′ ends, unlike siRNAs in *D. melanogaster*. Consistent with siRNA production by a processive Dicer-2 enzyme, Hi5 cells produce phased siRNAs from the RNA genome of a latent alphanodavirus. The commonalities and differences between *T. ni* and *D. melanogaster* small RNA pathways will help identify both deeply conserved and rapidly evolving components.

A major motivation for sequencing the *T. ni* genome was the establishment of a tractable cell culture model for studying small RNAs, especially piRNAs. We believe that our genome assembly and gene-editing protocols will enable the use of *T. ni* Hi5 cells to advance our understanding of how piRNA precursors are defined, made into piRNAs and act to silence transposons in the germline. Hi5 cells express essentially all known piRNA pathway genes except those specific to Drosophilids.

Furthermore, *T. ni* Vasa localizes to a perinuclear, nuage-like structure in Hi5 cells, making them suitable for studying the assembly of the subcellular structures thought to organize piRNA biogenesis. We have defined genomic piRNA-producing loci in Hi5 cells, as well as in the soma, testis, and ovary. The most productive piRNA clusters are shared among ovary, testis, and Hi5 cells. In addition, Hi5 cells contain novel piRNA clusters not found in the moth itself, suggesting that the process of establishing new piRNA-producing loci can be recapitulated by experimental manipulation of Hi5 cells.

As in *D. melanogaster*, splicing of *T. ni* piRNA precursor transcripts is efficiently suppressed, yet *T. ni* lacks paralogs of the proteins implicated in splicing suppression in flies. The ability to study the mechanisms by which piRNA clusters form and how precursor RNAs are transcribed, exported, and marked for piRNA production in *T. ni* promises to reveal both conserved and lepidopteran-specific features of this pathway. Notably, the W chromosome not only is a major piRNA source, but also produces piRNAs from almost its entirety. Future studies are needed to determine whether this is a common feature of W chromosomes in Lepidoptera and other insects.

The establishment of procedures for genome editing and single-cell cloning of Hi5 cells, combined with the *T. ni* genome sequence, make this germ cell line a powerful tool to study RNA and protein function ex vivo. Our strategy combines transfection of pre-assembled Cas9/sgRNA complexes with single clone isolation using a selectable marker (e.g. EGFP) and feeder cells physically separated from the engineered cells. Compared with nucleic-acid-based delivery of Cas9, transfection of Cas9 RNP minimizes the off-target mutations caused by prolonged Cas9 expression and eliminates the risk of integration of sgRNA or Cas9 sequences into the genome (*Lin et al., 2014*; *Kim et al., 2014*). Compared to plasmid donors (*Yu et al., 2014*; *Ge et al., 2016*), ssDNA homology donors similarly reduce the chance of introducing exogenous sequences at unintended genomic sites. Techniques for injecting the embryos of other lepidopteran species have already been established (*Wang et al., 2013*; *Takasu et al., 2014*; *Zhang et al., 2015*). In principle, Cas9 RNP injected into cabbage looper embryos could be used to generate genetically modified *T. ni* strains both to explore lepidopteran biology and to implement novel strategies for safe and effective pest control.

# Materials and methods

**Key resources table**

| Reagent type (species) or resource | Designation | Source or reference | Identifiers | Additional information |
|---|---|---|---|---|
| gene (*Trichoplusia ni*) | *vasa* | this paper | TNI000568 | |
| gene (*T. ni*) | *ciwi* | this paper | TNI008009 | |
| biological sample (*T. ni*) | Somatic tissue | Benzon Research | | male pupa |
| biological sample (*T. ni*) | Somatic tissue | Benzon Research | | female pupa |
| biological sample (*T. ni*) | Thorax | Benzon Research | | male adult |
| biological sample (*T. ni*) | Testes | Benzon Research | | male adult |
| biological sample (*T. ni*) | Thorax | Benzon Research | | female adult |
| biological sample (*T. ni*) | Ovaries | Benzon Research | | female adult |
| cell line (*T. ni*) | High Five (BTI-TN-5B1-4) | Thermo Fisher | Thermo Fisher: B85502 | wild type cell line |
| cell line (*T. ni*) | EGFP-HA-Vasa | this paper | | polyclonal stable cell line |
| cell line (*T. ni*) | Ciwi-mCherry | this paper | | monoclonal stable cell line |
| recombinant protein | EnGen Cas9 NLS | New England Biolabs | New England Biolabs: M0646T | |

*Continued on next page*

*Continued*

| Reagent type (species) or resource | Designation | Source or reference | Identifiers | Additional information |
|---|---|---|---|---|
| antibody | anti-GFP (mouse monoclonal) | Developmental Studies Hybridoma Bank | DSHB: DSHB-GFP-1D2; RRID:AB_2617419 | (1:200) |
| antibody | anti-HA (rabbit monoclonal) | Cell Signaling Technology | Cell Signaling Technology: 3724; RRID:AB_1549585 | (1:200) |
| antibody | Alexa Fluor 488-labeled donkey anti-mouse | Thermo Fisher | Thermo Fisher: A-21202 | (1:500) |
| antibody | Alexa Fluor 680-labeled donkey anti-rabbit | Thermo Fisher | Thermo Fisher: A10043 | (1:500) |
| recombinant DNA reagent | EGFP-HA-Vasa (linear dsDNA) | this paper | | synthesized gBlock from Integrated DNA technologies |
| sequence based reagents (DNA oligos) | GTTTTAGAGCTAGAAATAGCAAGTTAAAATAAGGCTAGTCCGTTATCAACTTGAAAAAGTGGCACCGAGTCGGTGC | this paper | tracr RNA Core | Used as template for sgRNA in vitro transcription |
| sequence based reagents (DNA oligos) | CATTTTGTGTTTCTCAACACTGG | this paper | sgRNA1 | sgRNA target site for *ciwi* deletion (PAM) |
| sequence based reagents (DNA oligos) | GGTACGGTGAGAAGCTCTACCGG | this paper | sgRNA2 | sgRNA target site for *ciwi* deletion (PAM) |
| sequence based reagents (DNA oligos) | GCTCAGTAGTAATAGATTTATGG | this paper | sgRNA3 | sgRNA target site for EGFP-HA-*vasa* mutation (PAM) |
| sequence based reagents (DNA oligos) | GGATGATGGTGTCGGTGATGTGG | this paper | sgRNA4 | sgRNA target site for EGFP-HA-*vasa* mutation (PAM) |
| sequence based reagents (DNA oligos) | ATGCTGCAGCTCCGGCGCGTAGG | this paper | sgRNA5 | sgRNA target site for mCherry-*ciwi* knockout (PAM) |
| sequence based reagents (DNA oligos) | TTTTCAATAACCCAAACATATGG | this paper | sgRNA6 | sgRNA target site for mCherry-*ciwi* knockout (PAM) |
| sequence based reagents (DNA oligos) | CtaatacgactcactataGGCATTTTGTGTTTCTCAACACgtttttagagct | this paper | T7-sgRNA1 forward primer | Forward primer for sgRNA in vitro transcription template generation |
| sequence based reagents (DNA oligos) | CtaatacgactcactataGGGGTACGGTGAGAAGCTCTACgtttttagagct | this paper | T7-sgRNA2 forward primer | Forward primer for sgRNA in vitro transcription template generation |
| sequence based reagents (DNA oligos) | CtaatacgactcactataGGGCTCAGTAGTAATAGATTTAgtttttagagct | this paper | T7-sgRNA3 forward primer | Forward primer for sgRNA in vitro transcription template generation |
| sequence based reagents (DNA oligos) | CtaatacgactcactataGGGGATGATGGTGTCGGTGATGgtttttagagct | this paper | T7-sgRNA4 forward primer | Forward primer for sgRNA in vitro transcription template generation |
| sequence based reagents (DNA oligos) | CtaatacgactcactataGGATGCTGCAGCTCCGGCGCGTgtttttagagct | this paper | T7-sgRNA5 forward primer | Forward primer for sgRNA in vitro transcription template generation |
| sequence based reagents (DNA oligos) | CtaatacgactcactataGGTTTTCAATAACCCAAACATAgtttttagagct | this paper | T7-sgRNA6 forward primer | Forward primer for sgRNA in vitro transcription template generation |
| sequence based reagents (DNA oligos) | GCACCGACTCGGTGCCACT | this paper | sgRNA reverse primer | Reverse primer for sgRNA in vitro transcription template generation |
| sequence based reagents (DNA oligos) | /Biotin/CGAATCGAAATCTAAGGCAAG | this paper | *vasa* donor forward | Forward primer for *vasa* HDR donor amplification |
| sequence based reagents (DNA oligos) | ATCTTTGGTGTGAGCTCAAGC | this paper | *vasa* donor reverse | Reverse primer for *vasa* HDR donor amplification |
| sequence based reagents (DNA oligos) | GCTATTTACCTACACAAACCAATTT | this paper | *ciwi* deletion forward | Forward primer for *ciwi* deletion detection |

*Continued on next page*

*Continued*

| Reagent type (species) or resource | Designation | Source or reference | Identifiers | Additional information |
|---|---|---|---|---|
| sequence based reagents (DNA oligos) | ACCACGACGTGATCCA | this paper | *ciwi* deletion reverse | Reverse primer for *ciwi* deletion detection |
| sequence based reagents (DNA oligos) | TGACTTGTGAATCCTTGGTTAC | this paper | vasa HR forward | Forward primer for *vasa* HR detection |
| sequence based reagents (DNA oligos) | CATTTTCATAATCCCTTGGTTCTC | this paper | vasa HR reverse | Reverse primer for *vasa* HR detection |
| sequence based reagents (DNA oligos) | GCGATAAATTGTTGGAAAC | this paper | GFP-HA-Vasa N-Fw | Forward primer for *vasa* HR insertion junction sequencing |
| sequence based reagents (DNA oligos) | TCATCCATCCCGCTAC | this paper | GFP-HA-Vasa N-Rv | Reverse primer for *vasa* HR insertion junction sequencing |
| sequence based reagents (DNA oligos) | GTTTAGAAACATGgtgagcaagg | this paper | GFP-HA-Vasa C-Fw | Forward primer for *vasa* HR insertion junction sequencing |
| sequence based reagents (DNA oligos) | CATTTTCATAATCCCTTGGTTCTC | this paper | GFP-HA-Vasa C-Rv | Reverse primer for *vasa* HR insertion junction sequencing |
| sequence based reagents (DNA oligos) | GTAAAACGACGGCCAG | this paper | M13 (-20) Fw | Forward primer for colony PCR |
| sequence based reagents (DNA oligos) | CAGGAAACAGCTATGAC | this paper | M13 Rv | Reverse primer for colony PCR |
| commercial kit | Express Five Serum Free Medium | Thermo Fisher | Thermo Fisher: 10486025 | Supplemented with 16mM L-Glutamine |
| commercial kit | NextSeq 500/550 High Output v2 kit (150 cycles) | Illumina | Illumina: FC-404-2005 | |
| commercial kit | NextSeq 500/550 High Output v2 kit (75 cycles) | Illumina | Illumina: FC-404-2002 | |
| commercial kit | Nextera Mate Pair Sample Prep Kit | Illumina | Illumina: FC-132-1001 | |
| commercial kit | TruSeq DNA LT Sample Prep Kit | Illumina | Illumina: FC-121-2001 | |
| commercial kit | Qubit dsDNA HS Assay kit | Thermo Fisher | Thermo Fisher: Q32851 | |
| commercial kit | SMRTbell Template Prep Kit 1.0 SPv3 | Pacific Biosciences | Pacific Biosciences: 100-991-900 | |
| commercial kit | ProLong Gold Antifade Mountant with DAPI | Thermo Fisher | Thermo Fisher: P36931 | |
| commercial kit | MirVana miRNA isolation kit | Thermo Fisher | Thermo Fisher: AM1561 | |
| commercial kit | Ribo-Zero Gold kit (Human/Mouse/Rat) | Epicentre | epicentre: MRZG12324 | |
| commercial kit | Trans-IT insect transfection reagent | Mirus Bio | Mirus Bio:MIR 6104 | |
| commercial kit | QIAquick Gel Extraction Kit | QIAGEN | QIAGEN:28704 | |
| commercial kit | Zero Blunt TOPO PCR Cloning Kit | Thermo Fisher | Thermo Fisher: K280020 | |
| commercial kit | M-280 streptavidin Dynabeads | Thermo Fisher | Thermo Fisher: 11205D | |
| software | online CRISPR design tool | http://crispr.mit.edu/ | PMID: 23873081 | |
| chemical compound | proteinase K | Sigma Aldrich | Sigma Aldrich: RPROTK-RO | |
| chemical compound | phenol:chloroform:isoamyl alcohol | Sigma Aldrich | Sigma Aldrich: P2069 | |

*Continued on next page*

*Continued*

| Reagent type (species) or resource | Designation | Source or reference | Identifiers | Additional information |
|---|---|---|---|---|
| chemical compound | RNase A | Sigma Aldrich | Sigma Aldrich: R4642 | |
| chemical compound | KaryoMAX Colcemid Solution in PBS | Life Technologies | Life Technologies: 15212012 | |
| chemical compound | Triton X-100 | Thermo Fisher | Thermo Fisher: NC1365296 | |
| chemical compound | PBS | Life Technologies | Life Technologies: 10010049 | |
| chemical compound | 16% formaldehyde | Thermo Fisher | Thermo Fisher: 28908 | |
| chemical compound | Photoflo 200 | Detek Inc | Detek Inc: 1464510 | |
| other | 22 x 22 mm cover slips | Thermo Fisher | Thermo Fisher:12541B | |
| other | 6-well plate | Corning | Corning: 351146 | |
| other | Transwell 96-well Receiver | Corning Life Sciences Plastic | Corning Life Sciences Plastic: 3382 | |
| Software | Canu v1.3 | doi:10.1101/gr.215087.116 | | |
| Software | LACHESIS | doi:10.1038/nbt.2727 | | |
| Software | BUSCO v3 | doi:10.1093/bioinformatics/btv351 | | |
| Software | piPipes | doi:10.1093/bioinformatics/btu647 | | |
| Software | MAKER | 10.1101/gr.6743907 | | |

## Genomic DNA libraries

Hi5 cells (ThermoFisher, Waltham, MA) were cultured at 27°C in Express Five Serum Free Medium (ThermoFisher) following the manufacturer's protocol. Thorax were dissected from four-day-old female or male *T. ni* pupa (Benzon Research, Carlisle, PA). Cells or tissues were lysed in 2 × PK buffer (200 mM Tris-HCl [pH7.5], 300 mM NaCl, 25 mM EDTA, 2% w/v SDS) containing 200 µg/ml proteinase K at 65°C for 1 hr, extracted with phenol:chloroform:isoamyl alcohol (25:24:1; Sigma, St. Louis, MO), and genomic DNA collected by ethanol precipitation. The precipitate was dissolved in 10 mM Tris-HCl (pH 8.0), 0.1 mM EDTA, treated with 20 µg/ml RNase A at 37°C for 30 min, extracted with phenol:chloroform:isoamyl alcohol (25:24:1), and collected by ethanol precipitation. DNA concentration was determined (Qubit dsDNA HS Assay, ThermoFisher). Genomic DNA libraries were prepared from 1 µg genomic DNA (Illumina TruSeq LT kit, NextSeq 500, Illumina, San Diego, CA).

Long-read genome sequencing with a 23 kb average insert range was constructed from 16 µg genomic DNA using the SMRTbell Template Prep Kit 1.0 SPv3 (Pacific Biosciences, Menlo Park, CA) according to manufacturer's protocol. Sequence analysis was performed using P6/C4 chemistry, 240 min data collection per SMRTcell on an RS II instrument (Pacific Biosciences). Mate pair libraries with 2 kb and 8 kb insert sizes were constructed (Nextera Mate Pair Library Prep Kit, Illumina) according to manufacturer's protocol from 1 µg Hi5 cell genomic DNA. Libraries were sequenced to obtain 79 nt paired-end reads (NextSeq500, Illumina).

## Hi-C

Hi-C libraries were generated from Hi5 cells as described (*Belton et al., 2012*), except that 50 million cells were used. Hi-C Libraries were sequenced using the NextSeq500 platform (Illumina) to obtain 79 nt, paired-end reads.

## Karyotyping

Hi5 cells were first incubated in Express Five medium containing 1 µg/ml colcemid at 27°C for 8 hr (*Schneider, 1979*), then in 4 ml 0.075 M KCl for 30 min at 37°C, and fixed with freshly prepared methanol:acetic acid (3:1, v/v) precooled to −20°C. Mitotic chromosomes were spread, mounted by incubation in ProLong Gold Antifade Mountant with DAPI (4′,6′-diamidino-2-phenylindole; Thermo-Fisher) overnight in the dark, and imaged using a DMi8 fluorescence microscope equipped with an 63 × 1.40 N.A. oil immersion objective (HCX PL APO CS2, Leica Microsystems, Buffalo Grove, IL) as described (*Matijasevic et al., 2008*).

## Small RNA libraries

Ovaries, testes, and thoraces were dissected from cabbage looper adults 24–48 hr after emerging. Total RNA (30 µg) was isolated (mirVana miRNA isolation kit, Ambion, Austin, TX) and sequenced using the NextSeq500 platform (Illumina) to obtain 59 nt single-end reads as previously described (*Han et al., 2015b*).

## RNA-seq

Adult ovaries, testes, or thoraces were dissected from cabbage looper adults 24 to 48 hr after emerging. Total RNA (3 µg) was purified (mirVana miRNA isolation kit, Ambion) and sequenced as described (*Zhang et al., 2012*) using the NextSeq500 platform (Illumina) to obtain 79 nt, paired-end reads.

## Genome assembly

Canu v1.3 (*Koren et al., 2017*) was used to assemble long reads into contigs, followed by Quiver (github.com/PacificBiosciences/GenomicConsensus) to polish the contigs using the same set of reads. Pilon (*Walker et al., 2014*) was used to further polish the assembly using Illumina paired-end reads. Finally, to assemble the genome into chromosome-length scaffolds, we joined the contigs using Hi-C reads and LACHESIS (*Burton et al., 2013*). The mitochondrial genome was assembled separately using MITObim (six iterations, *D. melanogaster* mitochondrial genome as bait; (*Hahn et al., 2013*).

To evaluate the quality of the genome assembly, we ran BUSCO v3 (*Simão et al., 2015*) using the arthropod profile and default parameters to identify universal single-copy orthologs. We further evaluated genome quality using conserved gene sets: OXPHOS and CRP genes. *B. mori* and *D. melanogaster* OXPHOS and CRP protein sequences were retrieved (*Marygold et al., 2007*; *Porcelli et al., 2007*) and BLASTp was used to search for their *T. ni* homologs, which were further validated by querying using InterPro (*Jones et al., 2014*; *Mitchell et al., 2015*). We also assembled *T. ni* genomes from male and female animals respectively using SOAPdenovo2 (kmer size 69; (*Luo et al., 2012*). We then compared the animal genomes with the *T. ni* genome assembled from Hi5 cells using QUAST (-m 500) (*Gurevich et al., 2013*) and the nucmer and mummerplot (–layout –filter) functions from MUMmer 3.23 (*Kurtz et al., 2004*). To determine the genomic variants, we used HaplotypeCaller from GATK (*McKenna et al., 2010*; *DePristo et al., 2011*; *Van der Auwera et al., 2013*) (-ploidy 4 -genotyping_mode DISCOVERY').

## Genome annotation

To annotate the *T. ni* genome, we first masked repetitive sequences and then integrated multiple sources of evidence to predict gene models. We used RepeatModeler to define repeat consensus sequences and RepeatMasker (-s -e ncbi) to mask repetitive regions (*Smit et al., 2017*). We used RNAmmer (*Lagesen et al., 2007*) to predict 8S, 18S, 28S rRNA genes, and Barrnap (https://github.com/tseemann/barrnap) to predict 5.8S rRNA genes. We used Augustus v3.2.2 (*Stanke et al., 2006*) and SNAP (*Korf, 2004*) to computationally predicted gene models. Predicted gene models were compiled by running six iterations of MAKER (*Campbell et al., 2014*), aided with homology evidence of well annotated genes (UniProtKB/Swiss-Prot and Ensembl) and of transcripts from related species (*B. mori* (*Suetsugu et al., 2013*) and *D. melanogaster* (*Attrill et al., 2016*). We used BLAST2GO (*Conesa et al., 2005*) to integrate results from BLAST, and InterPro (*Mitchell et al., 2015*) to assign GO terms to each gene. We used MITOS (*Bernt et al., 2013*) web server to predict mitochondrial genes and WebApollo (*Lee et al., 2013*) for manual curation of genes of interest. To

characterize telomeres, we used (TTAGG)$_{200}$ (*Robertson and Gordon, 2006*) as the query to search the *T. ni* genome using BLASTn with the option '-dust no' and kept hits longer than 100 nt. The genomic coordinates of these hits were extended by 10 kb to obtain the subtelomeric region.

## Orthology and evolution

To place genes into ortholog groups, we compared the predicted proteomes from 21 species (*Supplementary file 5*). Orthology assignment was determined using OrthoMCL (*Hirose and Manley, 1997*) with default parameters. MUSCLE v3.8.31 (*Edgar, 2004*) was used for strict 1:1:1 orthologs (*n* = 381) to produce sequence alignments. Conserved blocks (66,044 amino acids in total) of these alignments were extracted using Gblocks v0.91b (*Castresana, 2000*) with default parameters, and fed into PhyML 3.0 (*Vastenhouw et al., 2010*) (maximum likelihood, bootstrap value set to 1000) to calculate a phylogenetic tree. The human and mouse predicted proteomes were used as an outgroup to root the tree. The tree was viewed using FigTree (http://tree.bio.ed.ac.uk/software/figtree/) and iTOL (*Shirayama et al., 2012*).

## Sex determination and sex chromosomes

To identify sex-linked contigs, we mapped genomic sequence reads from males and females to the contigs. Reads with MAPQ scores $\geq$ 20 were used to calculate contig coverage, which was then normalized by the median coverage. The distribution of normalized contig coverage ratios (male:female ratios, M:F ratios) was manually checked to empirically determine the thresholds for Z-linked and W-linked contigs (M:F ratio >1.5 for Z-linked contigs and M:F ratio <0.5 for W-linked contigs). Lepidopteran *masc* genes were obtained from Lepbase (*Challis et al., 2016*). Z/AA ratio was calculated according to (*Gu et al., 2017*).

## Gene families for detoxification and chemoreception

To curate genes related to detoxification and chemoreception, we obtained seed alignments from Pfam (*Finn et al., 2016*) and ran hmmbuild to build HMM profiles of cytochrome P450 (P450), amino- and carboxy-termini of glutathione-*S*-transferase (GST), carboxylesterase (COE), ATP-binding cassette transporter (ABCs), olfactory receptor (OR), gustatory receptor (GR), ionotropic receptor (IR), and odorant binding (OBP) proteins, (*Supplementary file 6*, *7* and *8*). We then used these HMM profiles to search for gene models in the predicted *T. ni* proteome (hmmsearch, e-value cutoff: $1 \times 10^{-5}$). We also retrieved reference sequences of P450, GST, COE, ABC, OR, GR, IR, OBP, and juvenile hormone pathway genes from the literature (*Hekmat-Scafe et al., 2002*; *Bellés et al., 2005*; *Wanner and Robertson, 2008*; *Yu et al., 2008*; *Benton et al., 2009*; *Gong et al., 2009*; *Yu et al., 2009*; *Croset et al., 2010*; *Ai et al., 2011*; *Liu et al., 2011b*; *Dermauw and Van Leeuwen, 2014*; *Goodman and Granger, 2005*; *van Schooten et al., 2016*). These were aligned to the *T. ni* genome using tBLASTx (*Altschul et al., 1990*) and Exonerate (*Slater and Birney, 2005*) to search for homologs. Hits were manually inspected to ensure compatibility with RNA-seq data, predicted gene models, known protein domains (using CDD (*Marchler-Bauer et al., 2015*) and homologs from other species. P450 genes were submitted to David Nelson's Cytochrome P450 Homepage (*Nelson, 2009*) for nomenclature and classification. Sequences and statistics of these genes are in *Supplementary files 6*, *7* and *8*.

To determine the phylogeny of these gene families, we aligned the putative protein sequences from *T. ni* and *B. mori* genomes using MUSCLE (*Edgar, 2004*), trimmed the multiple sequence alignments using TrimAl (*Capella-Gutiérrez et al., 2009*) (with the option -automated1), and performed phylogenetic analysis (PhyML 3.0 (*Vastenhouw et al., 2010*), with parameters: -q –datatype aa –run_id 0 –no_memory_check -b −2). Phylogenetic trees were visualized using FigTree (http://tree.bio.ed.ac.uk/software/figtree/).

To curate opsin genes, we used opsin mRNA and peptide sequences from other species (*Zimyanin et al., 2008*; *Futahashi et al., 2015*) to search for homologs in *T. ni*. To discriminate opsin genes from other G-protein-coupled receptors, we required that the top hit in the NCBI non-redundant database and UniProt were opsins.

## Transposon analysis

To determine transposon age, we calculated the average percent divergence for each transposon family: the percent divergence (RepeatMasker) of each transposon copy was multiplied by its length, and the sum of all copies were divided by the sum of lengths of all copies in the family (*Pace and Feschotte, 2007*). We used TEMP (*Zhuang et al., 2014*) to identify transposon insertions in the Hi5 genome.

## miRNA and siRNA analysis

mirDeep2 (*Friedländer et al., 2008*, *2012*) with default parameters predicted miRNA genes. Predicted miRNA hairpins were required to have homology (exact seed matches and BLASTn e-value $<1 \times 10^{-5}$) to known miRNAs and/or miRDeep2 scores $\geq$ 10. miRNAs were named according to exact seed matches and high sequence identities (BLASTn e-value $<1 \times 10^{-5}$) with known miRNA hairpins. To determine the conservation status of *T. ni* miRNAs, putative *T. ni* miRNAs were compared with annotated miRNAs from *A. aegypti*, *A. mellifera*, *B. mori*, *D. melanogaster*, *H. sapiens*, *M. musculus*, *M. sexta*, *P. xylostella*, and *T. castaneum*: conserved miRNAs were required to have homologous miRNAs beyond Lepidoptera.

To compare siRNA abundance in oxidized and unoxidized small RNA-seq libraries, we normalized siRNA read counts to piRNA cluster-mapping reads (piRNA cluster read counts had >0.98 Pearson correlation coefficients between oxidized and unoxidized libraries in all cases). Because piRNA degradation products can be 20–22 nt long, we excluded potential siRNA species that were prefixes of piRNAs (23–35 nt).

To search for viral transcripts in *T. ni*, we downloaded viral protein sequences from NCBI (http://www.ncbi.nlm.nih.gov/genome/viruses/) and used using tBLASTn to map them to the *T. ni* genome and to the transcriptomes of Hi5 cells and five *T. ni* tissues. We filtered hits (percent identity $\geq$0.80, e-val $\leq 1 \times 10^{-20}$, and alignment length $\geq$100) and mapped small RNA-seq reads to the identified viral transcripts.

Candidate genomic hairpins were defined according to *Okamura et al. (2008b)*). Candidate *cis*-NATs were defined according to (*Ghildiyal et al., 2008*).

## piRNA analysis

To determine the genomic coordinates of piRNA-producing loci, we mapped small RNAs to the genome as described (*Han et al., 2015a*). We then calculated the abundance of piRNAs in 5 kb genomic windows. For each window, we counted the number of uniquely mapped reads and the number of reads mapped to multiple loci (multimappers) by assigning reads using an expectation-maximization algorithm. Briefly, each window had the same initial weight. The weight was used to linearly apportion multimappers. During the expectation (E) step, uniquely mapped reads were unambiguously assigned to genomic windows; multimappers were apportioned to the genomic windows they mapped to, according to the weights of these windows. At the maximization (M) step, window weights were updated to reflect the number of reads each window contained from the E step. The E and M steps were run iteratively until the Manhattan distance between two consecutive iterations was smaller than 0.1% of the total number of reads.

To identify differentially expressed piRNA loci, we used the ppm and rpkm values, normalized to the total number of uniquely mapped reads, to measure piRNA abundance. For analyses including all mapped reads (uniquely mapped reads and multimappers), reads were apportioned by the number of times that they were mapped to the genome. To make piRNA loci comparable across tissues, we merged piRNA loci from ovary, testis, female and male thorax, and Hi5 cells. For the comparison between female and male thoraces, the cluster on tig00001980 was removed as this cluster likely corresponds to a mis-assembly. We used Spearman correlations to calculate the pairwise correlations of piRNA abundances. As for defining sex-linked contigs, we calculated M:F ratios and used the same thresholds to determine whether a piRNA cluster was sex-linked. A piRNA locus was considered to be differentially expressed if the ratio between the two tissues was >2 or<0.5 and FDR < 0.1 (after t-test).

Splice sites were deemed to be supported by RNA-seq data when supported by at least one data set. We used AUGUSTUS (*Stanke et al., 2006*), with the model trained for *T. ni* genome-wide gene prediction, to predict gene models and their splice sites in *T. ni* piRNA clusters.

## β-elimination

Total RNAs were extracted from Hi5 cells using mirVana kit as described previously. We then incubated 100 μg total RNA with 25 mM NaIO$_4$ in borate buffer (148 mM Borax,148 mM Boric acid, pH 8.6) for 30 min at room temperature, beta-elimination was performed in 50 mM NaOH at 45°C for 90 min (*Horwich et al., 2007*). The resultant RNA was collected by ethanol precipitation.

## sgRNA design

sgRNAs for the target loci (5′-end of *TnPiwi* and 5′-end of *vasa*) were designed using crispr.mit.edu (*Hsu et al., 2013*) to retrieve all possible guide sequences, and guide sequences adjacent to deletion or insertion targets were chosen. *Supplementary file 9* lists guide sequences.

## ssDNA donor purification

Donor template sequence was produced as a gBlock (Integrated DNA Technologies, San Diego, CA). A biotinylated forward primer and a standard reverse primer were used in PCR to generate a double-stranded, biotinylated DNA donor. The biotinylated DNA was captured on M-280 streptavidin Dynabeads (ThermoFisher), and the biotinylated strand was separated from the non-biotinylated strand essentially as described in the manufacturer's protocol. *Supplemental file 10* provides a detailed protocol.

## Transfection of Hi5 cells

sgRNAs were transcribed using T7 RNA polymerase, gel purified, then incubated with Cas9 in serum-free Hi5 culture medium supplemented with 18 mM ʟ-glutamine. The resulting sgRNA/Cas9 RNPs were incubated with Trans-IT insect reagent (Mirus Bio, Madison, WI) for 15 min at room temperature, then evenly distributed onto 90% confluent Hi5 cells. Culture medium was replaced with fresh medium 12 hr later. Genomic DNA was isolated and analyzed by PCR 48 hr later.

## PCR to validate genomic editing in transfected cells

Forty eight hours after transfection, Hi5 cells from one 90% confluent well of a six-well plate (Corning, Corning, NY) were collected, washed once with PBS (ThermoFisher) and lysed in 2 × PK buffer containing 200 μg/ml proteinase K, extracted with phenol:chloroform:isoamyl alcohol (25:24:1), and then genomic DNA collected by ethanol precipitation. Deletions in *TnPiwi* were detected by PCR using primers flanking the deleted region (*Supplementary file 9*). To confirm deletions by sequencing PCR products were resolved by agarose gel electrophoresis, purified (QIAquick Gel Extraction Kit, QIAGEN, Germantown, MD, USA), and cloned into pCR-Blunt II-Topo vector (ThermoFisher). The recombinant plasmid was transformed into Top10 competent *E.coli* (ThermoFisher) following supplier's protocol. PCR products amplified using M13 (−20) forward and M13 reverse primers from a sample of a single bacterial colony were sequenced by GENEWIZ (South Plainfield, NJ).

## Single clone selection

Wild-type Hi5 cells were seeded into a 96-well Transwell permeable support receiver plate (Corning, Corning, NY) at 30% confluence and incubated overnight in serum free medium with 100 U/ml penicillin and 100 μg/ml streptomycin. A Transwell permeable support insert plate with media in each well was inserted into the receiver plate, and a single EGFP-positive cell was sorted into each insert well by FACS. After 14 days incubation at 27°C, wells were examined for EGFP-positive cell clones using a DMi8 fluorescent microscope (Leica).

## Immunostaining

EGFP-HA-Vasa-expressing Hi5 cells were seeded on 22 × 22 mm cover slips (Fisher Scientific, Pittsburgh, PA) in a well of a six-well plate (Corning). After cells had attached to the coverslip, the medium was removed and cells were washed three times with PBS (Gibco). Cells were fixed in 4% (w/v) methanol-free formaldehyde (ThermoFisher) in PBS at room temperature for 15 min, washed three times with PBS, permeabilized with 0.1% (w/v) Triton X-100 in PBS for 15 min at room temperature, and then washed three times with PBS. For antibody labeling, cells were incubated in 0.4% (v/v) Photo-Flo in 1 × PBS for 10 min at room temperature, then 10 min in 0.1% (w/v) Triton X-100 in PBS and 10 min in 1 × ADB PBS (3 mg/ml bovine serum albumen, 1% (v/v) donkey serum, 0.005%

(w/v) Triton X-100 in 1 × PBS). Next, cells were incubated with primary antibodies (mouse anti-GFP antibody (GFP-1D2, Developmental Studies Hybridoma Bank, Iowa City, IA) and rabbit anti-HA Tag antibody (C29F4, Cell Signaling, Danvers, MA), diluted 1:200 in ADB (30 mg/ml BSA, 10% (v/v) donkey serum, 0.05% (w/v) Triton X-100 in 1 × PBS) at 4°C overnight. After three washes in PBS, cells were incubated sequentially in 0.4% (v/v) Photo-Flo in 1 × PBS, 0.1% (w/v) Triton X-100 in PBS, and 1 × ADB PBS, each for 10 min at room temperature. Cells were then incubated with secondary Alexa Fluor 488-labeled donkey anti-mouse (ThermoFisher) and Alexa Fluor 680-labeled donkey anti-rabbit (ThermoFisher) antibodies, diluted 1:500 in ADB at room temperature for one hour. After washing three times with 0.4% (v/v) Photo-Flo in 1 × PBS and once with 0.4% (v/v) Photo-Flo in water, coverslips were air dried in the dark at room temperature. Slides were mounted in ProLong Gold Antifade Mountant with DAPI and examined by confocal microscopy (TCS SP5 II Laser Scanning Confocal, Leica).

## Data deposition

The *T. ni* Whole Genome Shotgun project has been deposited at DDBJ/ENA/GenBank under the accession NKQN00000000. The version described here is version NKQN01000000. All sequencing data are available through the NCBI Sequence Read Archive under the accession number PRJNA336361. Further details are available at the Cabbage Looper Database (http://cabbagelooper.org/).

## Acknowledgements

We thank members of the Weng and Zamore laboratories for helpful discussions and comments on the manuscript; UMass Deep Sequencing Core for Pacific Biosciences sequencing; Zdenka Matijasevic for sharing the karyotyping protocol. This work was supported in part by National Institutes of Health grants R37GM062862 to PDZ and HD078253 to ZW.

## Additional information

### Funding

| Funder | Grant reference number | Author |
|---|---|---|
| National Institutes of Health | HD078253 | Zhiping Weng |
| Howard Hughes Medical Institute | | Phillip D Zamore |
| National Institutes of Health | R37GM062862 | Phillip D Zamore |

The funders had no role in study design, data collection and interpretation, or the decision to submit the work for publication.

### Author contributions

Yu Fu, Data curation, Formal analysis, Investigation, Methodology, Writing—original draft, Writing—review and editing; Yujing Yang, Investigation, Methodology, Writing—original draft, Writing—review and editing; Han Zhang, Gwen Farley, Investigation, Methodology; Junling Wang, Investigation, DNA-seq; Kaycee A Quarles, Investigation, P. xylostella small RNA-seq; Zhiping Weng, Supervision, Funding acquisition, Methodology, Writing—original draft, Project administration, Writing—review and editing; Phillip D Zamore, Conceptualization, Supervision, Funding acquisition, Visualization, Methodology, Writing—original draft, Project administration, Writing—review and editing

### Author ORCIDs

Yu Fu http://orcid.org/0000-0003-1244-9473
Han Zhang http://orcid.org/0000-0003-1090-2967
Phillip D Zamore http://orcid.org/0000-0002-4505-9618

Decision letter and Author response
Decision letter https://doi.org/10.7554/eLife.31628.040
Author response https://doi.org/10.7554/eLife.31628.041

## Additional files

### Supplementary files

• Supplementary file 1. *T. ni* genome statistics. (A) BUSCO assessments of *T. ni* and six other genomes. (B) CRP genes. (C) Genes in the OXPHOS pathway. (D) Genome comparisons. Genomes assembled using paired-end DNA-seq data from male and female *T. ni* pupae are compared with the Hi5 genome as the reference. The dot plots show genome alignments for contigs $\geq$ 1 kb. (E) Numbers of genes in lepidopteran genomes. (F) Positions of telomeric repeats: position of $(TTAGG)_n$ longer than 100 nt. (G) Transposons in *T. ni* subtelomeric regions. (H) Repeat statistics for the *T. ni* genome. (I) Transposon family divergence rates. (J) Manual curation of W-linked protein-coding genes and miRNAs.
DOI: https://doi.org/10.7554/eLife.31628.022

• Supplementary file 2. Genes encoding small RNA pathway proteins. (A) Genes encoding miRNA and siRNA pathway proteins. (B) Genes encoding piRNA pathway proteins (grouped by sequence orthology).
DOI: https://doi.org/10.7554/eLife.31628.023

• Supplementary file 3. *T. ni* miRNAs, siRNAs and piRNAs. (A) miRNA annotation. (B) Mapping statistics for endogenous siRNAs in *T. ni* and *D. melanogaster*. (C) piRNA cluster lengths. piRNA cluster coordinates in Hi5 (D), ovary (E), testis (F), female thorax (G), and male thorax (H).
DOI: https://doi.org/10.7554/eLife.31628.024

• Supplementary file 4. mirDeep2 output for *T. ni* miRNAs
DOI: https://doi.org/10.7554/eLife.31628.025

• Supplementary file 5. Genomes used in this study.
DOI: https://doi.org/10.7554/eLife.31628.026

• Supplementary file 6. *T. ni* detoxification-related genes. (A) P450 gene counts by clade in *T. ni* and *B. mori*. (B) Sequences of P450 proteins. (C) Sequences of glutathione-*S*-transferase proteins. (D) Carboxylesterase gene counts by clade in *T. ni* and *B. mori*. (E) Sequences of carboxylesterase proteins. (F) ATP-binding cassette transporter gene counts by clade in *T. ni* and *B. mori*. (G) Sequences of ATP-binding cassette transporter proteins.
DOI: https://doi.org/10.7554/eLife.31628.027

• Supplementary file 7. *T. ni* chemoreception genes. (A) Sequences of olfactory receptor proteins. (B) Sequences of gustatory receptor proteins. (C) Sequences of ionotropic receptor proteins.
DOI: https://doi.org/10.7554/eLife.31628.028

• Supplementary file 8. Genes in the juvenile hormone biosynthesis and degradation pathways.
DOI: https://doi.org/10.7554/eLife.31628.029

• Supplementary file 9. Genome-modified sequences.
DOI: https://doi.org/10.7554/eLife.31628.030

• Supplementary file 10. Single-stranded DNA donor purification
DOI: https://doi.org/10.7554/eLife.31628.031

• Transparent reporting form
DOI: https://doi.org/10.7554/eLife.31628.032

### Major datasets

The following datasets were generated:

| Author(s) | Year | Dataset title | Dataset URL | Database, license, and accessibility information |
|---|---|---|---|---|
| Yu Fu, Yujing Yang, Han Zhang, Gwen Farley, Junling Wang, Kaycee A Quarles, Zhiping Weng, Phillip D Zamore | 2018 | *Trichoplusia ni* isolate ovarian cell line Hi5, whole genome shotgun sequencing project | https://www.ncbi.nlm.nih.gov/nuccore/NKQN00000000 | Publicly available at NCBI Nucleotide (accession no. NKQN00000000) |
| Yu Fu, Yujing Yang, Han Zhang, Gwen Farley, Junling Wang, Kaycee A Quarles, Zhiping Weng, Phillip D Zamore | 2018 | The genome of the Hi5 germ cell line from *Trichoplusia ni*, an agricultural pest and novel model for small RNA biology | https://www.ncbi.nlm.nih.gov/bioproject/PRJNA336361 | Publically avaialble at NCBI BioProject (accession no. PRJNA336361) |

The following previously published dataset was used:

| Author(s) | Year | Dataset title | Dataset URL | Database, license, and accessibility information |
|---|---|---|---|---|
| Megha Ghildiyal, Jia Xu, Herve Seitz, Zhiping Weng, Phillip D Zamore | 2010 | Sorting of Drosophila small silencing RNAs partitions microRNA* strands into the RNA interference pathway | https://www.ncbi.nlm.nih.gov/geo/query/acc.cgi?acc=GSE18806 | Publically avaialble at NCBI Gene Expression Omnibus (accession no: GSE18806) |

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
