## [Decision Letter]

Thank you for submitting your article "The genome of *Trichoplusia ni*, an agricultural pest and novel model for small RNA biology" for consideration by *eLife*. Your article has been reviewed by two peer reviewers, and the evaluation has been overseen by a Reviewing Editor and Diethard Tautz as the Senior Editor. The following individual involved in review of your submission has agreed to reveal his identity: Julius Brennecke (Reviewer #1).

The reviewers have discussed the reviews with one another and the Reviewing Editor has drafted this decision to help you prepare a revised submission.

The paper presents a draft assembly of the genome of the destructive lepidopteran insect pest, *Trichoplusia ni*, and it derives a number of interesting and important features of both the coding and non-coding parts of the genome.

There is a relatively (to the second part of the paper) brief description of gene orthology based on a comparison with an Arthropoda dataset with reference to the opsin gene family, genes associated with sex determination and an analysis of Z chromosome dosage compensation associated with sex determination. There is novelty in the W chromosome sequence that is not available for other Lepidoptera although its enrichment for repeats and gene depletion is apparently similar in other members of that order.

The structural features of the genome (telomeres, absence of well-defined centromeres, heterochromatin) are well documented as are the transposons and repeats.

Added to this standard genome analysis there is a lengthy description of the small RNAome including the miRNAs, viral and other siRNAs (that are surprisingly not 2'-O methylated and piRNAs and the piRNA proteins. The *T. ni* pathway is revealed as a variation on other piRNA pathways including the presence of both dual and uni-strand piRNA clusters, a large cluster accounting for most if not all of the W chromosome. This description is coupled to a lengthy description of the expression pattern of the various piRNA-producing regions, including suppression of splicing. There is also a description of genome editing coupled to single cloning of mutant cultured cells in the *Pi* pathway *ciwi* gene and a transient assay system showing that the piRNA pathway protein Vasa is in perinuclear nuages, as in Drosphila.

Required Revisions:

1) Figure 1, Table1 and Supplementary file 1 genome assembly statistics, are the numbers of reads and fold coverage relating to the sequencing from mainly Hi5 cells genomic DNA, or do they also include sequencing reads from the male and female pupae? The authors note the very fragmented assemblies and short N50 from the pupae genomic DNA, and the impression received is that these reads were mainly useful in determining which of the scaffolds correspond to W, Z sex chromosomes versus the autosomes. Although the vast majority of the pupae reads map to the *T. ni* genome assembly, were they sufficient to confirm that the higher-level scaffold order within the de-novo assembled chromosomes in the Hi5 cell genome is the same as in *T. ni* pupae? We think this distinction is important, and the text and paper title should clarify that this high-quality genome assembly is mainly based off the *T. ni* Hi5 cell genome. It would be useful to comment on whether the Hi5 cell for the genome assembly was derived from expansion of a single clone, and if not, how does their pipeline handle the tetraploidy/karyotype variability from a mixed cell population?

2) Can the authors comment on whether *T. ni* miRNAs vs. siRNAs also partition into which of the two Ago/Dcr proteins like in *Drosophila*? When *T. ni* Ago2/Dcr2 are compared to *Drosophila* Ago2/Dcr2, are there differences in the alignment that would yield insight into lack of 2OMe in the siRNAs, perhaps differences in the Ago2 PAZ domain?

3) The authors claim that the 'Entire' W chromosome is a giant piRNA cluster, but is every base on W truly covered by sequenced piRNAs? Are all the few protein-coding genes on W also generating piRNAs? Since many miRNA loci were also mapped to W, are these loci also generating piRNAs at the same time? If not, then perhaps revise as "nearly the entire W chromosome".

4) How thorough is the CRISPR knockout of *ciwi* – do the modified/selected Hi5 cells show piRNA depletion despite the presence of the remaining unmodified genomic copy in Figure 7? How stable is this modification in subsequent propagation of the Ciwi-modified hi5 cells? The image of the mCherry-tagged Ciwi in Figure 7—figure supplement 1 is very fuzzy and unlike the sharper images of the eGFP-tagged Vasa cells. Have the authors confirmed by genomic PCR and sequencing that the mCherry is inserted into the Ciwi locus like their data showing eGFP inserted into the Vasa locus? The concern stems from the literature describing efficient DNA repair/recombination mechanisms in lepidopteran cells like BmN4, and this may lend to challenges in genome editing. Could the tetraploidy of Hi5 cells cause issues in achieving complete knock out of genes with Cas9 and single-cell cloning? Can RNAi be performed to knockdown gene expression in Hi5 cells like BmN4 cells, which the Kirano and Siomi labs can do with a prolonged and repeated dsRNA treatment protocol?

5) The authors nicely show TNCL virus-mapping siRNAs, but are there also TNCL piRNAs, such as virus-derived piRNAs observed in mosquito cells? Are there piRNA clusters that map to the 3'UTR of protein-coding genes as seen in other animals?

6) We understand the nomenclature history for Piwi genes using animal colloquial names (mouse *miwi*, human *hiwi*, chicken *chiwi*) but one may worry that *ciwi* cabbage looper *piwi* will be confused with cat *piwi*, cow *piwi*, and camel *piwi*. We suggest *TnPiwi*, the more scalable naming convention that the authors used for TnAgo3 (like BmAgo3, DmAgo3).

7) In the second paragraph of the Introduction and in the second paragraph of the subsection “Genome-editing and single-cell cloning of Hi5 cells”, the authors state BmN4 cells are difficult to grow, but I believe this statement is incorrect because many labs have used these cells, and the Tomari lab can grow sufficient BmN4 cells to purify Siwi for piRNA biochemical studies, so I would presume these cells are straightforward to grow. Please comment.

[Editors' note: further revisions were requested prior to acceptance, as described below.]

Thank you for resubmitting your work entitled "The genome of the Hi5 germ cell line from *Trichoplusia ni*, an agricultural pest and novel model for small RNA biology" for further consideration at *eLife*. Your revised article has been favorably evaluated by Diethard Tautz (Senior Editor), a Reviewing Editor, and two reviewers.

The manuscript has been improved but there are some remaining issues that need to be addressed before acceptance, as outlined below:

1) This revision still includes this Figure 7—figure supplement 1 and text in the fifth paragraph of the subsection “Genome-editing and single-cell cloning of Hi5 cells”. The mCherry data is also tied to the text describing single-cell cloning and Figure 8. If the authors remove the mCherry data, how will this impact the description of single-cell cloning, and the manuscript revision is still unclear if the EGFP-HA-vasa cells were isolated via this single-cell cloning? The text says EGFP-positive cells are detected one week after Cas9/sgRNA/ssDNA transfection, but single cell sorting/growth requires at least 2 weeks, and there is no detail on whether EGFP-HA-Vasa required this 3-week regimen?

2) From author rebuttal: "We have demonstrated that we can make genomic deletions, we have not yet knocked out all four copies of a gene. The most likely explanation for this is that in the absence of all Ago3 or Ciwi, Hi5 cells are inviable. We are actively working to develop protocols to knockout all four copies of a gene, but these methods will take time to test."

This explanation would be valuable to include in the manuscript itself, either in Results or in the Discussion, to clarify why the WT band remains in the δ lane of Figure 7. Also in Figure 7, the prime characters in 5' and 3' are not rendering correctly in this revision PDF or print?

3) Citations are still missing in the first paragraph of the Introduction [cauliflower, prolonged culture] and in the first paragraph of the subsection “Genome sequencing and assembly” [tetraploid]. Also, additional EndNote formatting issues in the subsection “piRNA pathway proteins”, #77489?

4) We are glad the authors agree to using TnPiwi, and with regards to the rebuttal, we also note the challenge of pronouncing "*ciwi*" differently from Bombyx Siwi (not /k/iwi?). Nevertheless, this raises another question as to why in Supplementary file 2 TnPiwi is ascribed to the *Drosophila* homolog Aubergine rather than Piwi, if the authors note they do not know if TnPiwi functions more like Aub or Piwi? Is Supplementary file 2 reflecting more closely related protein sequence between TnPiwi and Aub versus Piwi? TnAub?

---

## [Author Response]

Required Revisions:1) Figure 1, Table1 and Supplementary file 1 genome assembly statistics, are the numbers of reads and fold coverage relating to the sequencing from mainly Hi5 cells genomic DNA, or do they also include sequencing reads from the male and female pupae? The authors note the very fragmented assemblies and short N50 from the pupae genomic DNA, and the impression received is that these reads were mainly useful in determining which of the scaffolds correspond to W, Z sex chromosomes versus the autosomes. Although the vast majority of the pupae reads map to the T. ni genome assembly, were they sufficient to confirm that the higher-level scaffold order within the de-novo assembled chromosomes in the Hi5 cell genome is the same as in T.ni pupae? We think this distinction is important, and the text and paper title should clarify that this high-quality genome assembly is mainly based off the T. ni Hi5 cell genome. It would be useful to comment on whether the Hi5 cell for the genome assembly was derived from expansion of a single clone, and if not, how does their pipeline handle the tetraploidy/karyotype variability from a mixed cell population?

The numbers in Figure 1, Table 1, and Supplementary file 1 come from the genome sequence assembled using data from Hi5 cells. We now make this clearer in the figure and table legends. Wild-caught insect populations typically have high levels of heterozygosity, so it would have been difficult if not impossible to obtain a highly contiguous genome sequence using the insect itself. To our knowledge, no lepidopteran species has ever been successfully inbred, and our attempts to inbreed *T. ni* predictably failed due to inbreed depression between generations five and nine.

With our current datasets, we cannot confirm that the higher-level scaffold order is the same between Hi5 cells and *T. ni*.

While we did not re-derive our Hi5 cell line from a single clone – because methods to clone these cells were only developed in our lab after the genome sequence was completed, the original Hi5 cell line presumably derives from a single cell. To test this supposition, we identified variants in the Hi5 genome sequence. In total, we called variants at 165,370 genomic positions (0.0449% of the reference genome). For the majority of these genomic positions (88.8%, covering 0.0399% of the genome), only one copy of the chromosome has the variant allele while the other three chromosomal copies match the reference genome. We can make three conclusions. First, Hi5 cells originated from a single founder cell or a homogenous population of cells. Second, the founder cells were haploid, consistent with the fact that Hi5 cells were isolated from eggs not ovaries. Third, most sequence variants were acquired after the original derivation of the line from *T. ni* eggs.

Such a low level of heterozygosity – one heterozygous site every ~2,200 bases – can be readily handled by short read aligners such as Bowtie and BWA and by the assembler Canu. For the overwhelming majority of genomic positions (>99.9%), these tetraploid cells have four identical copies of chromosomes.

2) Can the authors comment on whether T. ni miRNAs vs. siRNAs also partition into which of the two Ago/Dcr proteins like in Drosophila? When T.ni Ago2/Dcr2 are compared to Drosophila Ago2/Dcr2, are there differences in the alignment that would yield insight into lack of 2OMe in the siRNAs, perhaps differences in the Ago2 PAZ domain?

We have no empirical data that speak to how miRNAs and siRNAs partition in *T. ni.* We would like to note that in our unpublished, ongoing work, we find that many other insects similarly do not 2′-*O*-methylate their siRNAs. We aligned the fly and *T. ni* protein sequences for Dcr-2 and Ago2. The fly and cabbage looper proteins are considerably different throughout the entire protein, preventing us from speculating on what is different between the orthologs.

3) The authors claim that the 'Entire' W chromosome is a giant piRNA cluster, but is every base on W truly covered by sequenced piRNAs? Are all the few protein-coding genes on W also generating piRNAs? Since many miRNA loci were also mapped to W, are these loci also generating piRNAs at the same time? If not, then perhaps revise as "nearly the entire W chromosome".

We should have qualified “entire,” as the reviewer correctly points out. We have not only revised the text, but included a more thoughtful analysis of our data. Many of the protein-coding genes on the W are likely transposons or transposon fragments. These produce abundant piRNAs, while genes with true protein-coding potential produced far fewer piRNAs. In contrast, the miRNA loci on the W are unlikely to be real, because (1) they are unique to *T. ni*, and (2) show a ping-pong signature. We now suggest these may be piRNA-producing loci and not authentic miRNAs.

4) How thorough is the CRISPR knockout of ciwi – do the modified/selected Hi5 cells show piRNA depletion despite the presence of the remaining unmodified genomic copy in Figure 7?

We have not yet performed those experiments – which lie outside the scope of our current manuscript – because we are working to obtain a full series of knockouts (disruption of 1, 2, 3, or all 4 copies of *ciwi*).

How stable is this modification in subsequent propagation of the Ciwi-modified hi5 cells?

Understanding the stability of knockouts of piRNA pathway genes is a long-term project, beyond the scope of the proof-of-principle experiments in our manuscript. We are now comparing several strategies, including drug selection, to examine this, but we cannot see the utility of further delaying the manuscript – likely by many months.

The image of the mCherry-tagged Ciwi in Figure 7—figure supplement 1 is very fuzzy and unlike the sharper images of the eGFP-tagged Vasa cells.

The image is fuzzy because it is a photograph of the single-cell clone growing on a flexible membrane above a layer of feeder cells. It isn’t technically possible for us to get a sharper image of a single-cell clone prior expanding it in a traditional plastic cell culture dish.

Have the authors confirmed by genomic PCR and sequencing that the mCherry is inserted into the Ciwi locus like their data showing eGFP inserted into the Vasa locus?

Yes, the insertion of mCherry into *ciwi* was confirmed by sequencing of a genomic PCR product.

The concern stems from the literature describing efficient DNA repair/recombination mechanisms in lepidopteran cells like BmN4, and this may lend to challenges in genome editing. Could the tetraploidy of Hi5 cells cause issues in achieving complete knock out of genes with Cas9 and single-cell cloning? Can RNAi be performed to knockdown gene expression in Hi5 cells like BmN4 cells, which the Kirano and Siomi labs can do with a prolonged and repeated dsRNA treatment protocol?

Our efforts to date have focused on knock-ins like mCherry-tagged Ciwi or the eGFPtagged Vasa. While we have demonstrated that we can make genomic deletions, we have not yet knocked out all four copies of a gene. The most likely explanation for this is that in the absence of all Ago3 or Ciwi, Hi5 cells are inviable. We will, but have not yet tested whether RNAi can be performed in Hi5 cells. Thoroughly testing and optimizing Cas9 and RNAi protocols is beyond the scope of this manuscript.

5) The authors nicely show TNCL virus-mapping siRNAs, but are there also TNCL piRNAs, such as virus-derived piRNAs observed in mosquito cells? Are there piRNA clusters that map to the 3'UTR of protein-coding genes as seen in other animals?

It is unlikely that there are TNCL^-^mapping piRNAs. We have added this sentence, “The TNCL^-^mapping small RNAs include some 23–32 nt RNAs. These are unlikely to be antiviral piRNAs, because they lack the characteristic first-nucleotide uridine bias and show no significant ping-pong signal (*Z*-score = −0.491). We conclude that Hi5 cells do not use piRNAs for viral defense.”

Although we can identify 3′ UTR-mapping piRNAs (2.0%–3.8% of piRNAs in *T. ni* tissues and Hi5 cells), manual inspection of the genome sequence suggests that many 3′ UTRs have been missed or have little RNA-seq support. This most likely reflects the poor performance of ab initio prediction of UTRs in non-model organisms. For example, the AUGUSTUS (one of the gene predictors used in this study) manual explicitly mentions that its UTR prediction is limited to a few species. MAKER (the genome annotation pipeline used in this study) can use EST evidence to update UTR features, but this is limited to genes producing abundant transcripts. Thus, the current state of automated genome annotation does not permit us to confidently draw conclusions about 3′ UTR piRNAs. Sequencing RNAs in more tissues coupled with large-scale manual curation would be required to speak to the issue of 3′ UTR-mapping piRNAs, tasks clearly beyond the scope of the current manuscript.

6) We understand the nomenclature history for Piwi genes using animal colloquial names (mouse miwi, human hiwi, chicken chiwi) but one may worry that ciwi cabbage looper piwi will be confused with cat piwi, cow piwi, and camel piwi. We suggest TnPiwi, the more scalable naming convention that the authors used for TnAgo3 (like BmAgo3, DmAgo3).

We agree, begrudgingly, and have changed Ciwi to TnPiwi throughout the manuscript. We note that to avoid confusion with *ciwi*, cat *piwi* could have been *meowi*, cow *piwi, moowi*, and camel *piwi, humpwi*.

7) In the second paragraph of the Introduction and in the second paragraph of the subsection “Genome-editing and single-cell cloning of Hi5 cells”, the authors state BmN4 cells are difficult to grow, but I believe this statement is incorrect because many labs have used these cells, and the Tomari lab can grow sufficient BmN4 cells to purify Siwi for piRNA biochemical studies, so I would presume these cells are straightforward to grow. Please comment.

Our experience was that BmN4 cells are difficult to grow compared to other insect and mammalian cell lines we have used, but to gain more insight into this question, we reached out to Dr. Tomari: “Growing BmN4 cells is not super easy but practically straightforward to grow; they are a bit more tricky than S2 cells (especially when starting from the frozen stock) but not so bad as OSC (we gave up growing OSC in our hands).” In light of his reply, we no longer comment on the difficulty in growing BmN4 cells, and we have revised the text to read, “*T. ni* Hi5 cells grow rapidly without added hemolymph, are readily transfected, and – unlike *B. mori* BmN4 cells, which also express germline piRNAs – remain homogeneously undifferentiated even after prolonged culture,” and “Unfortunately, BmN4 cells readily differentiate into two morphologically distinct cell types,”.

[Editors' note: further revisions were requested prior to acceptance, as described below.]

The manuscript has been improved but there are some remaining issues that need to be addressed before acceptance, as outlined below:1) This revision still includes this Figure 7—figure supplement 1 and text in the fifth paragraph of the subsection “Genome-editing and single-cell cloning of Hi5 cells”.

We have removed Figure 7—figure supplement 1 and the corresponding text.

The mCherry data is also tied to the text describing single-cell cloning and Figure 8. If the authors remove the mCherry data, how will this impact the description of single-cell cloning, and the manuscript revision is still unclear if the EGFP-HA-vasa cells were isolated via this single-cell cloning?

While the manuscript was being reviewed, the EGFP-HA-Vasa-expressing cells were cloned by the same procedure. We have replaced the image in Figure 8 with one of the cells after single-cell cloning and have made this more clear in the text.

The text says EGFP-positive cells are detected one week after Cas9/sgRNA/ssDNA transfection, but single cell sorting/growth requires at least 2 weeks, and there is no detail on whether EGFP-HA-Vasa required this 3-week regimen?

The EGFP-positive EGFP-HA-Vasa cells have now been cloned by the same procedure, and the text rewritten to reflect this.

2) From author rebuttal: "We have demonstrated that we can make genomic deletions, we have not yet knocked out all four copies of a gene. The most likely explanation for this is that in the absence of all Ago3 or Ciwi, Hi5 cells are inviable. We are actively working to develop protocols to knockout all four copies of a gene, but these methods will take time to test."This explanation would be valuable to include in the manuscript itself, either in Results or in the Discussion, to clarify why the WT band remains in the δ lane of Figure 7.

We have added this explanation to the text: “We note that these cells still contain at least one wild-type copy of *TnPiwi*. We have not yet obtained cells in which all four copies of *TnPiwi* are disrupted, perhaps because in the absence of Piwi, Hi5 cells are inviable.”

Also in Figure 7, the prime characters in 5' and 3' are not rendering correctly in this revision PDF or print?

We are not sure how to fix this, as the prime characters render correctly at our end. We will work with the copy editor to ensure that they are correct in the final published version.

3) Citations are still missing in the first paragraph of the Introduction [cauliflower, prolonged culture] and in the first paragraph of the subsection “Genome sequencing and assembly” [tetraploid]. Also, additional EndNote formatting issues in the subsection “piRNA pathway proteins”, #77489?

We have fixed these issues.

4) We are glad the authors agree to using TnPiwi, and with regards to the rebuttal, we also note the challenge of pronouncing "ciwi" differently from Bombyx Siwi (not /k/iwi?). Nevertheless, this raises another question as to why in Supplementary file 2 TnPiwi is ascribed to the Drosophila homolog Aubergine rather than Piwi, if the authors note they do not know if TnPiwi functions more like Aub or Piwi? Is Supplementary file 2 reflecting more closely related protein sequence between TnPiwi and Aub versus Piwi? TnAub?

As the reviewer points out, we did not use functional homology to classify the piRNA pathway genes. Instead, they are classified by sequence orthology (now noted in Supplementary file 2). By amino acid identity, TnPiwi is closer to Aub than to Piwi. For example, when TnPiwi is used as query for a BLASTP search of the NCBI non-redundant protein database, fly Aub ranks higher (44% sequence identity) than Piwi (40% sequence identity). A similar conclusion was reached by Xiol et al. (*Cell* 2014). Because the name “Aubergine” is restricted to flies, where for historical reasons it describes a developmental phenotype rather than involvement in the piRNA pathway, we have retained the more widely used name, Piwi, from which piRNAs take their name.